# Macrophage migration inhibitory factor is overproduced through EGR1 in *TET2*[low] resting monocytes

Elodie Pronier[1,2,16], Aygun Imanci[1,3,16], Dorothée Selimoglu-Buet [1,3], Bouchra Badaoui[4], Raphael Itzykson [5], Thierry Roger [6], Chloé Jego[1,3], Audrey Naimo[7], Maëla Francillette[7], Marie Breckler[7], Orianne Wagner-Ballon [4,8], Maria E. Figueroa [9,10], Marine Aglave[11], Daniel Gautheret[11], Françoise Porteu[1,3], Olivier A. Bernard[3,12], William Vainchenker [1,3], François Delhommeau[1,13,14], Eric Solary [1,3,15] & Nathalie M. Droin [1,3,7✉]

Somatic mutation in *TET2* gene is one of the most common clonal genetic events detected in age-related clonal hematopoiesis as well as in chronic myelomonocytic leukemia (CMML). In addition to being a pre-malignant state, *TET2* mutated clones are associated with an increased risk of death from cardiovascular disease, which could involve cytokine/chemokine overproduction by monocytic cells. Here, we show in mice and in human cells that, in the absence of any inflammatory challenge, *TET2* downregulation promotes the production of MIF (macrophage migration inhibitory factor), a pivotal mediator of atherosclerotic lesion formation. In healthy monocytes, TET2 is recruited to *MIF* promoter and interacts with the transcription factor EGR1 and histone deacetylases. Disruption of these interactions as a consequence of *TET2*-decreased expression favors EGR1-driven transcription of *MIF* gene and its secretion. MIF favors monocytic differentiation of myeloid progenitors. These results designate MIF as a chronically overproduced chemokine and a potential therapeutic target in patients with clonal *TET2* downregulation in myeloid cells.

[1] INSERM U1287, Gustave Roussy Cancer Center, 94805 Villejuif, France. [2] Owkin Lab, Owkin, Inc., New York, NY 10003, USA. [3] Université Paris Saclay, Faculté de Médecine, 94270 Le Kremlin-Bicêtre, France. [4] AP-HP, Hôpitaux Universitaires Henri-Mondor, Département d'Hématologie et Immunologie Biologiques, 94000 Créteil, France. [5] AP-HP, Service Hématologie Adultes, Hôpital Saint-Louis, 75010 Paris, France. [6] Infectious Disease Service, Department of Medicine, Centre Hospitalier Universitaire Vaudois and University of Lausanne, 1011 Lausanne, Switzerland. [7] INSERM US23, CNRS UMS 3655, AMMICa, Genomic platform, Gustave Roussy Cancer Center, 94805 Villejuif, France. [8] Université Paris Est Créteil, INSERM, IMRB, Equipe 9, 94010 Créteil, France. [9] Human Genetics, University of Miami Miller School of Medicine, 33136 Miami, USA. [10] Sylvester Comprehensive Cancer Center, University of Miami Miller School of Medicine, 33136 Miami, USA. [11] INSERM US23, CNRS UMS 3655, AMMICa, Bioinformatic platform, Gustave Roussy Cancer Center, 94805 Villejuif, France. [12] INSERM U1170, Gustave Roussy Cancer Center, 94805 Villejuif, France. [13] Sorbonne Université, Inserm, Centre de Recherche Saint-Antoine, CRSA, 75012 Paris, France. [14] AP-HP, Sorbonne Université, Hôpital Saint-Antoine, Service d'Hématologie et Immunologie Biologique, 75012 Paris, France. [15] Hematology department, Gustave Roussy Cancer Center, 94805 Villejuif, France. [16]These authors contributed equally: Elodie Pronier, Aygun Imanci. ✉email: nathalie.droin@gustaveroussy.fr

Ten-eleven-translocation (TET) proteins are iron [Fe(II)]- and α-ketoglutarate (α-KG)-dependent dioxygenases that promote active DNA demethylation through iterative oxidation of 5-methylcytosine (5mC) to 5-hydroxymethylcytosine (5hmC), 5-formylcytosine (5fC), and 5-carboxylcytosine (5caC), eventually leading to the replacement of 5mC by native C[1]. hmC also prevents DNA methylation by decreasing cytosine accessibility to DNA methyltransferases[2]. This activity of TET enzymes is regulated via substrate and cofactor availability, post-transcriptional regulation, and post-translational modifications[3].

TET family member interacting proteins may tether them to DNA. For example, TET2 could interact with NANOG in mouse embryonic stem cells[4], SPI1/PU.1 in differentiating B-cells[5] and monocytes[6], early growth response 2 (EGR2) in IL4/GM-CSF-driven human monocyte differentiation[7] and Wilms' tumor suppressor gene1 (WT1) in acute myeloid leukemia cells[8,9]. The binding of TET proteins to 5mC-free promoters with diverse partners suggested that they could function independently of their catalytic activity[3]. In brain epigenome programming during postnatal development, EGR1 recruits TET1 to demethylate EGR1 binding sites[10]. Tet1 regulates gene transcription in mouse embryonic stem cells through associating with the Sin3A co-repressor complex[11] and MOF histone acetyltransferase[12], while a catalytically dead Tet2 mutant represses *Interleukin-6* (*Il6*) gene transcription in mouse macrophages by recruiting Hdac1 and Hdac2 histone deacetylases[13]. Some of the catalytic-activity independent effects of TET proteins are mediated by the recruitment of OGT [O-linked N-acetylglucosamine (O-GlcNAc) transferase] to gene promoters[14–16].

Mono- or bi-allelic somatic mutations along the entire coding TET2 region are recurrent events in human hematopoietic malignancies[17,18], especially in chronic myelomonocytic leukemia (CMML) in which mono- or bi-allelic mutations in TET2 gene are detected in 57% of patients[19]. Mouse models with Tet2 gene deletion in hematopoietic stem cells develop a myeloid or a lymphoid malignancy[20–23]. The long latency and low penetrance of these diseases suggest that cooperation with another genetic event and/or a permissive environment may be needed for malignancy emergence[24–26].

Somatic mutation in TET2 gene is also one of the most common clonal genetic events detected in the peripheral blood of ageing healthy individuals, named Clonal Hematopoiesis of Indeterminate Potential (CHIP)[27]. These TET2-mutated clones can be a first step towards a malignancy such as CMML[28]. A TET2-mutated CHIP also increases the risk of death from cardiovascular disease[29,30]. A deregulated production of inflammatory cytokines by mutated myeloid cells may explain the cardiovascular risk associated with TET2 CHIP as these cytokines may promote leukocyte recruitment to atherosclerotic plaques[31,32]. Accordingly, Tet2-deletion in murine macrophages induces a constitutive expression of lipopolysaccharide (LPS)-induced genes[33] and cardiovascular risk is reduced on a genetically decreased IL-6 signaling background[34] or prevented by a selective NLR family pyrin domain-containing 3 (NLRP3) inflammasome inhibitor decreasing IL-1β secretion by innate immune cells[35].

Here, we show that, in the absence of any inflammatory challenge, TET2 gene downregulation induces an overproduction and secretion of MIF (macrophage migration inhibitory factor). Initially identified as a lymphocyte-derived soluble product[36,37], MIF is released by a variety of cells[38] and behaves as a proinflammatory cytokine with pathogenic roles in inflammatory and autoimmune disorders[39–44]. Genetic deletion of *Mif* impairs the production of inflammatory mediators by monocytes/macrophages[45,46] and prevents the inflammasome activation[47]. We show that in CMML human monocytes harboring truncating variants of TET2 gene, MIF secretion is increased through EGR-1 transcription factor recruitment to its promoter. These results identify MIF as a potential therapeutic target to prevent atherosclerosis and progression of TET2 mutated CHIP towards a chronic myeloid malignancy.

## Results

**TET2 downregulation induces MIF overproduction**. We investigated whether TET2 gene downregulation could alter cytokine secretion by myeloid cells. Human cord blood CD34+ cells, transduced with TET2 or scrambled (SCR) shRNA lentiviruses, were sorted and cultured with SCF, IL-3, TPO, and GM-CSF to promote granulocytic/monocytic differentiation[48]. Using cytokine-arrays to analyze day-10 cell culture supernatant, three cytokines were readily detected: MIF, G-CSF, and IL-1RA. MIF was repeatedly increased, while G-CSF and IL-1RA were not affected when TET2 expression was decreased (Fig. 1a, b). ELISA measurements confirmed MIF overproduction upon TET2 silencing at days 8–10 of culture (Fig. 1c). Consistent with these observations, MIF mRNA (Fig. 1d) and secreted MIF (Fig. 1e, f) were increased in four human leukemic cell lines (kasumi-1, M07e, UT-7, and TF-1) in which TET2 gene expression was decreased by using lentiviral shRNA (Supplementary Fig. 1a), as previously described[48]. Finally, MIF concentrations were increased in blood (Fig. 1g) and supernatant of bone marrow aspirations (Supplementary Fig. 1b) of two Tet2-deficient mouse models[20]. In these models, Mif plasma levels increased in mice 1–3 aged months, i.e., before changes in white blood cell count (Supplementary Fig. 1c, d), thus monocyte counts in the peripheral blood as a confounding factor. These results indicated that, in vitro and in vivo, in mouse and human cells, TET2 downregulation increased MIF secretion.

**MIF is overexpressed in TET2-mutated CMML monocytes**. CMML is a clonal disorder, mostly observed in the elderly with a median age at diagnosis of 73 years. TET2 mutations are early somatic events[49] and the most frequent genetic alteration of this disease[50]. To explore the link between TET2 mutations and MIF expression in human primary samples, we sorted peripheral blood monocytes from 60 CMML patients (17 TET2-wild-type [$TET2^{WT}$] and 43 TET2-mutated [$TET2^{MUT}$] cases) (Supplementary Table 1) and from 10 age-matched healthy donors and performed bulk RNA sequencing. Focusing on cytokine genes, IL6 mRNA was not expressed in control and CMML resting monocytes while a significant increase in MIF mRNA was observed in $TET2^{MUT}$ compared to healthy donors and $TET2^{WT}$ CMML (Fig. 2a; Supplementary Data 1). We also observed an inverse correlation between TET2 and MIF mRNA expression levels in CMML patient monocytes (Fig. 2b). MIF-increased mRNA detected in $TET2^{MUT}$ monocytes was validated by RT-qPCR analysis in an independent cohort of 146 CMML patients (56 $TET2^{WT}$ and 90 $TET2^{MUT}$ cases including 68 truncating variants) compared to 19 young and 8 age-matched healthy donors (Supplementary Table 2 and Fig. 2c). Importantly, we did not detect any significant recurrent change in the expression of other cytokine-encoding genes in TET2-mutated patient monocytes, i.e., MIF gene expression was frequently increased in cells expressing truncating TET2 variant whereas the increased expression of other cytokine-encoding genes was heterogeneous and independent of TET2 status (Fig. 2d).

In a multi-institutional study (n = 1084 CMML patients), we recently reported an overall survival advantage associated with TET2 mutations in CMML patients. This was especially significant in the context of multiple or truncating TET2 mutants ($trTET2^{MUT}$), i.e., nonsense and frameshift variants, compared to

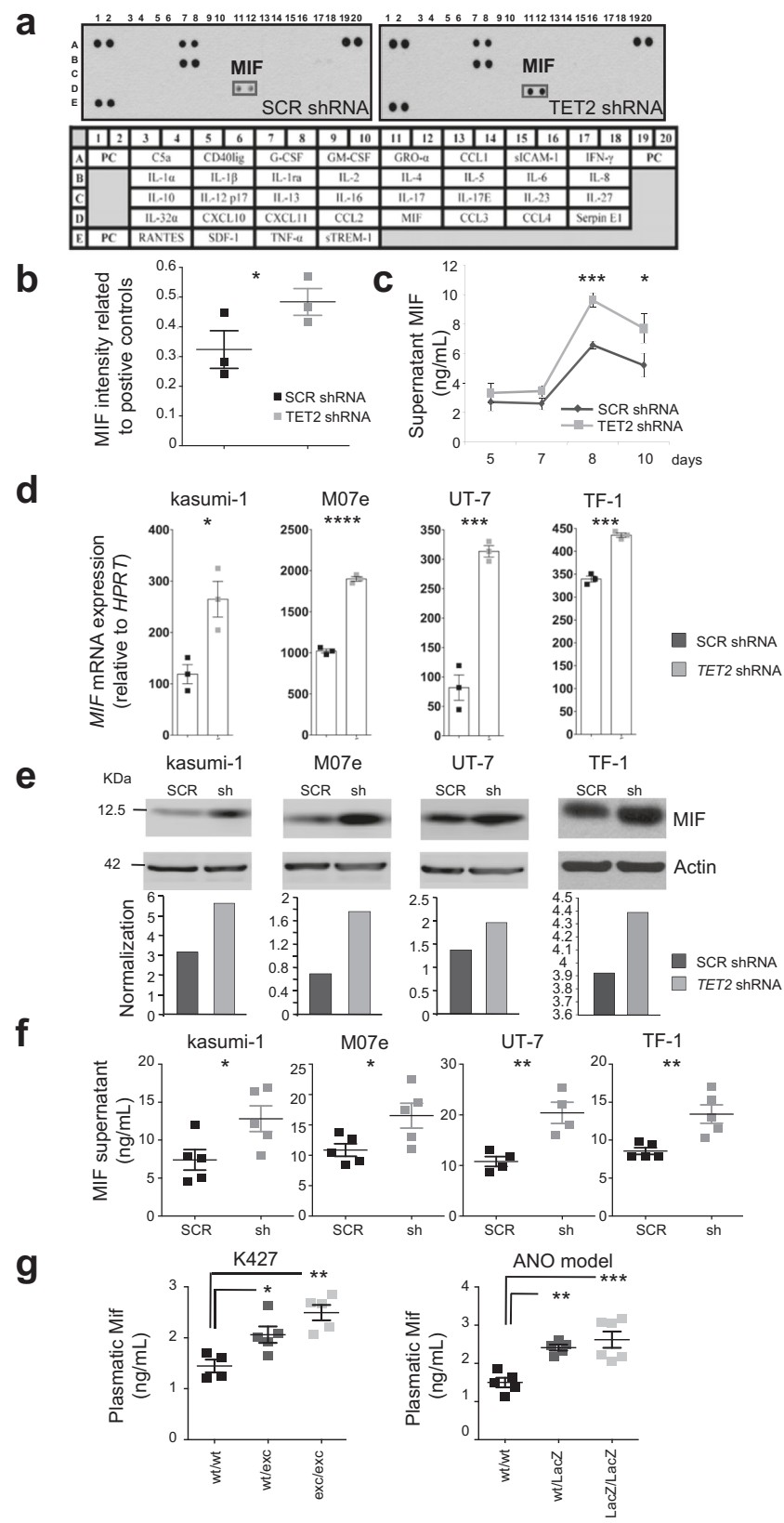

non-truncated variants (*non-trTET2^MUT*), i.e., in frame insertion/deletion, missense, and splice site variants[19]. From our RNA sequencing data, the decreased expression of *TET2* gene expression observed in *trTET2^MUT* cells correlated with an increased expression of *MIF* mRNA, which was not detected in

*non-trTET2^MUT* cases (Fig. 2e). Among CMML samples tested by RT-qPCR (Fig. 2c), we observed an increased expression of *MIF* gene in *trTET2^MUT* samples only (Fig. 2f). The previously described better survival of *TET2^MUT* CMML patients was validated in the present cohort (Supplementary Fig. 2a) with no

**Fig. 1 *TET2* downregulation promotes MIF production. a, b** Cytokine profile arrays of supernatant collected from cord blood CD34$^+$ cells infected with *SCR*- and *TET2*-shRNA-GFP lentiviruses, sorted on GFP expression, and induced to differentiate with stem cell factor (SCF), interleukin-3 (IL-3), Fms-related tyrosine kinase 3 ligand (FLT3L) and granulocyte-colony stimulating factor (G-CSF). **a** Representative cytokine array with supernatants collected at day 10 of differentiation. The rectangle points to MIF detection. **b** Quantification of MIF signals, normalized to positive controls. Data are mean $+/-$ SEM of three independent experiments. Paired t test: *$P < 0.05$. **c** MIF concentrations determined by ELISA in the supernatant of cells induced to differentiate for indicated time. Data are mean $+/-$ SEM of indicated independent experiments (day 5: $n = 6$; day 7: $n = 5$; day 8: $n = 3$; day 10: $n = 7$). Paired t test: *$P < 0.05$; ***$P < 0.001$. **d** RT-qPCR analysis of *MIF* mRNA expression in four *TET2*-depleted (*TET2* shRNA, gray bars) and control (*SCR* shRNA, black bars) human leukemic cell lines. Data are mean $+/-$ SEM of three biological replicates. Unpaired t test: *$P < 0.05$; ***$P < 0.001$; ****$P < 0.0001$. **e** Immunoblot of *SCR* or *TET2* shRNA infected leukemic cell lines sorted on GFP expression. Lower panels, quantification after actin normalization using Image J software. **f** MIF concentrations determined by ELISA in the supernatants of kasumi-1 ($n = 5$), M07e ($n = 5$), UT-7 ($n = 4$) and TF-1 ($n = 5$) cells transduced 24 h before with SCR (black squares) or TET2 (gray squares) shRNA. Data are mean $+/-$ SEM of indicated biological replicates. Unpaired-t test: *$P < 0.05$; **$P < 0.01$. **g** MIF concentrations determined by ELISA in the plasma of two *Tet2*-deficient models (1–3 months): K427 knock-out model (wt/wt or *Tet2*$^{+/+}$ $n = 4$, wt/exc or *Tet2*$^{+/-}$ $n = 5$ and exc/exc or *Tet2*$^{-/-}$ $n = 5$); ANO knock-down model (wt/wt or *Tet2*$^{+/+}$ $n = 5$, wt/LacZ or *Tet2*$^{+/-}$ $n = 5$ and LacZ/LacZ or *Tet2*$^{-/-}$ $n = 6$). Data are mean $+/-$ SEM of indicated biological replicates. Dunnett's multiple comparison tests using wt/wt as control: *$P < 0.05$, **$P < 0.01$, ***$P < 0.001$.

significant difference between *trTET2*$^{MUT}$ and *non-trTET2*$^{MUT}$ CMML (Supplementary Fig. 2b). No difference in white blood cell, monocyte, neutrophil, lymphocyte count, and hemoglobin level was observed between the three groups whereas, interestingly, the platelet count was significantly lower in *trTET2*$^{MUT}$ patient subgroup (Supplementary Fig. 2c).

Immunoblot analysis of monocytes from 3 controls and 5 CMML patients (1 *TET2*$^{WT}$, 4 *trTET2*$^{MUT}$) further showed an increased expression of MIF protein in *trTET2*$^{MUT}$ patients (Supplementary Fig. 3a). MIF level was higher in the supernatant (18 h in serum-free medium) of *trTET2*$^{MUT}$ CMML-monocytes from 2 patients compared to healthy donors (Supplementary Fig. 3b, c) and in the serum of 3 *trTET2*$^{MUT}$ CMML patients at diagnosis compared to 3 healthy donor serum samples (Supplementary Fig. 3b, d). Finally, MIF concentrations in the supernatant of bone marrow aspirates from 10 healthy controls and 35 CMML patients (13 *TET2*$^{WT}$, 19 *trTET2*$^{MUT}$, and 3 *non-trTET2*$^{MUT}$) (Supplementary Table 3) revealed a significant increase of MIF in *trTET2*$^{MUT}$ CMML samples (Fig. 2g). Together, these results identified a correlation between *TET2* mutation, especially truncated mutants, and increased production of MIF.

**TET2 protein binds *MIF* promoter region.** Tet2 chromatin immunoprecipitation-sequencing (ChIP-seq) performed with mouse wild-type bone-marrow cells[16] indicated the recruitment of Tet2 at the *Mif* promoter (chromosome 10, locus position 75,322,231–75,323,742 using the mouse genome NCBI Build 37/UCSC mm9). Using two distinct antibodies against TET2 (Ab1:C2 and Ab2: sc-136926), we performed ChIP-qPCR to map TET2 binding in 3 regions (R2-R4) spanning the *MIF* promoter (Fig. 3a) in control (shSCR) and *TET2*-depleted (shTET2) kasumi-1 cells. In control cells, TET2 was enriched around the transcription start site (TSS) (Fig. 3b). In both humans and mice, downregulation of *TET2* precluded chromatin immunoprecipitation, supporting the specificity of TET2 binding. We then analyzed H3K4me3 and H3K27me3 histone marks and recruitment of active polymerase II (phosphorylated on serine 5, pS5 Pol II) to the *MIF* promoter. H3K4me3 signal increased from the distal region of *MIF* promoter to the TSS region in shSCR cells, while it was decreased in shTET2 cells (Fig. 3c) as previously shown[16]. H3K27me3 was not detected at the *MIF* promoter (Fig. 3c). Finally, active polymerase II was recruited in the TSS area of the *MIF* gene at much higher levels in shTET2 than shSCR cells (Fig. 3d), in accordance with an increased transcription of *MIF* gene under *TET2* depletion.

To determine the minimal DNA sequences regulating *MIF* promoter activity, we used fragments of the human *MIF* promoter, ranging from position −1083 to +129, into a *luciferase* reporter vector that was transiently expressed into SCR- and TET2-shRNA kasumi-1 cells. The −25/+129 and +44/+129 vectors drove only background luciferase activity while the other vectors induced a stronger luciferase activity in *TET2*-depleted compared to control cells (Fig. 4a). Since the −81/+129 sequence was sufficient to drive optimal MIF promoter activity, we explored the role of cis-acting regulatory elements in that region. Deletion of c-MYB, CRE$^d$, SP1$^d$, AML1a, AP4, and HIF1 sites[51] did not affect the luciferase activity measured in SCR- and TET2-shRNA kasumi-1 cells. Deletion of the SP1$^p$ and CRE$^p$ sites decreased luciferase activity in both control and *TET2*-depleted cells, indicating that these sites are critical for *MIF* gene expression (Fig. 4b). Using the Transcription Element Search System (TESS, http://www.cbil.upenn.edu/tess/), we identified a potential EGR binding site overlapping the SP1$^p$ site (Fig. 4c). We noticed a higher recruitment of EGR1 to the R4 region of the *MIF* promoter in *TET2*-depleted cells, whereas EGR2 was only slightly more recruited and SP1 recruitment remained unchanged (Supplementary Fig. 4). Together, these results identify EGR1 recruitment to the minimal promoter sequence required for *MIF* gene expression.

**TET2 interacts with EGR1.** In silico analysis using MethylPrimer Express (Applied Biosystems) identified a CpG island (CG >65% of 710 bp) within MIF gene promoter. Sequencing of bisulfite-treated DNA failed to identify any differential methylation of the minimal promoter sequence (−81/+129) in a cohort of 21 CMML monocytes, including 11 TET2$^{WT}$ and 10 TET2$^{MUT}$ samples, compared to 17 healthy donors, including 10 young and 7 age-matched donors, respectively (Supplementary Fig. 5). We next performed ChIP-qPCR in controls, *TET2*$^{WT}$, and *TET2*$^{MUT}$ CMML patient monocytes, focusing on the R3 region, close to the TSS (Fig. 3a). EGR1 was strongly enriched in *trTET2*$^{MUT}$ cells compared to controls and *TET2*$^{WT}$ cells (Fig. 5a). To demonstrate EGR1-dependent *MIF* expression in *trTET2*$^{MUT}$ cells, we transfected EGR1-siRNA in *trTET2*$^{MUT}$ monocytes collected from 2 CMML patients. *EGR1* mRNA downregulation, validated by RT-qPCR (Fig. 5b and Supplementary Fig. 6a), prevented *MIF* upregulation in these cells. HDAC1 and HDAC2 were enriched on *MIF* promoter in controls and *TET2*$^{WT}$ cells, but their recruitment decreased in *trTET2*$^{MUT}$ cells (Fig. 5c). To determine if EGR1 and TET2 could interact, we transiently co-transfected 293T cells with empty vectors or pcDNA3-EGR1 and pcDNA3-TET2-HA. Having checked EGR1 and TET2-HA protein expression in 293T cells (Fig. 5d), co-immunoprecipitation experiments validated the ability of TET2 to interact with EGR1. Interestingly, HDAC1 was only immunoprecipitated with

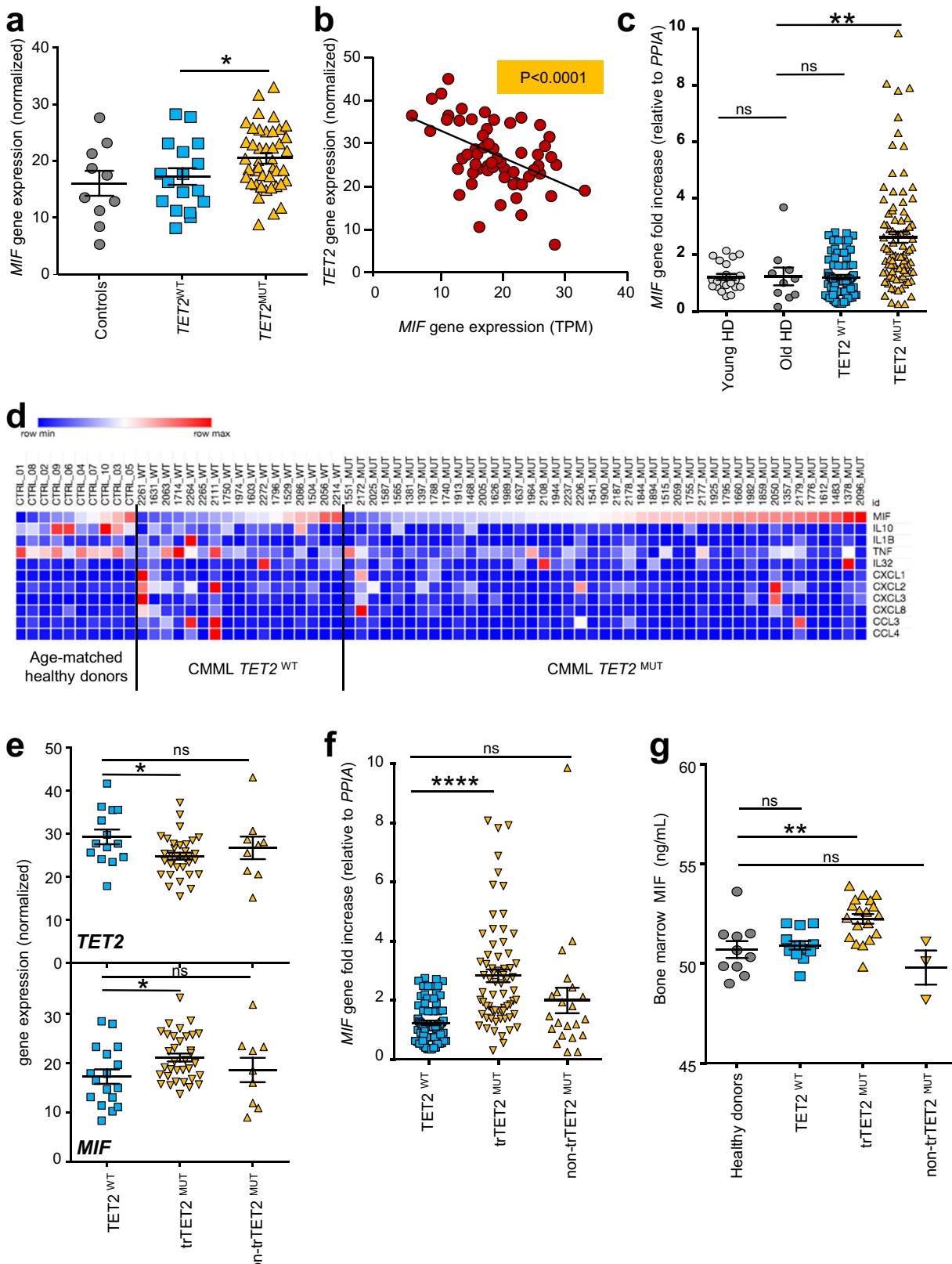

TET2-HA antibody, suggesting a stronger interaction with TET2 in the complex (Fig. 5d). Co-immunoprecipitation experiments in human blood monocytes confirmed TET2 interaction with EGR1 (Supplementary Fig. 6b, c) and validated the ability of TET2 to interact with HDAC1 and HDAC2 (Supplementary Fig. 6d).

**TET2 downregulation disrupts TET2/EGR1 interaction**. To explore if *TET2* downregulation globally affects EGR1 recruitment to DNA in monocytes, we performed ChIP-seq in monocytes from two healthy donors and three *TET2^MUT^* CMML patients selected for having *trTET2^MUT^* with diverse variant allele frequencies (VAF). CMML1818 combined *TET2* A1241fsX11

**Fig. 2 Increased expression of *MIF* gene in *TET2*-mutant CMML monocytes. a–c** Sequencing of total polyA-RNA was performed in sorted peripheral blood monocytes collected from age-matched healthy donors ($n = 10$, gray circles) and CMML patients ($n = 60$) including $TET2^{WT}$ ($n = 17$, blue squares) and $TET2^{MUT}$ ($n = 43$, orange triangles) cases. **a** *MIF* gene expression after count normalization. Data are mean $+/-$ SEM of indicated biological samples. Mann–Whitney test using $TET2^{WT}$ as control: \*$P < 0.05$. **b** Inverse correlation between *TET2* and *MIF* gene expression in all samples (10 age-matched healthy donors and 60 CMML patients); $R^2 = 0.2325$; $P < 0.0001$. **c** RT-qPCR analysis (normalized to *PPIA* gene) of *MIF* mRNA expression in sorted peripheral blood monocytes of 19 young healthy donors (age < 65), 8 age-matched healthy donors (age > 64) and 146 CMML patients ($TET2^{WT}$ 56; $TET2^{MUT}$ 90). Data are mean $+/-$ SEM of indicated biological samples. Dunnett's multiple comparison test using Old HD as control, \*\*$P < 0.01$, ns, non-significant. **d** Heatmap of the expression of a selection of cytokine and chemokine genes in the three cohorts. **e** *TET2* and *MIF* gene expression after count normalization was compared in CMML monocytes of patients with truncating $tr$-$TET2^{MUT}$ ($n = 34$) and non-truncating ($n = 9$) $non$-$trTET2^{MUT}$. Data are mean $+/-$ SEM of indicated biological samples. Dunnett's multiple comparison test using $TET2^{WT}$ ($n = 17$) as control, \*$P < 0.05$, ns, non-significant. **f** *MIF* gene expression normalized to *PPIA* gene was compared in CMML monocytes of patients with truncating $trTET2^{MUT}$ ($n = 68$) and non-truncating ($n = 22$) $non$-$trTET2$;$^{MUT}$ Data are mean $+/-$ SEM of indicated biological samples. Dunnett's multiple comparison test using $TET2^{WT}$ ($n = 56$) as control, \*\*\*\*$P < 0.0001$, ns, non-significant; **g** MIF concentrations in bone marrow fluid from healthy controls ($N = 10$), $TET2^{WT}$ ($n = 12$), $trTET2^{MUT}$ ($n = 19$) and $non$-$trTET2^{MUT}$ ($n = 3$) CMML patients. Data are mean $+/-$ SEM of indicated biological samples. Dunnett's multiple comparison test using Old HD as control, \*\*$P < 0.01$, ns non-significant.

(VAF = 40%) and E1513GfsX9 (VAF = 49%), CMML1900 exhibited a *TET2* W564X mutant (VAF = 100%) and CMML1268 showed a *TET2* H800fs variant (VAF = 16%). EGR1 recruitment was increased in CMML1818 and CMML1900 monocytes, both exhibiting high $trTET2^{MUT}$ allele frequencies, compared to CMML1268 and control monocytes (Fig. 6a). Altogether, Cis-Regulatory Annotation System (CEAS) indicated that EGR1 was preferentially recruited to intergenic regions in control and CMML1268 monocytes, whereas it was highly recruited at TSS in the two CMML samples with high $trTET2^{MUT}$ allele frequencies (Fig. 6b). Ranking heatmaps centered on TSS confirmed EGR1 distribution around the TSS in the two CMML samples with high $trTET2^{MUT}$ VAF (Fig. 6c). Examples of EGR1 distribution around TSS are shown for *RAD50* (Fig. 6d), *RPL30*, *RPL37A*, *RPS13*, *RPS23*, *RPS3A*, *POLR1B*, *POLR1D*, and *POLR1E* (Supplementary Fig. 7a). In control samples and CMML1268, EGR1 peaks were very similar and mainly intergenic, as illustrated for *CEBPD* (Fig. 6d), *ALDH3B2* and *BCL2* regions as well as intergenic areas on chromosome 2, 7, 9, 13, 16, and 22 (Supplementary Fig. 7b).

MACS2 algorithm identified 11,393 and 19,763 peaks in control samples, of which 2,069 (13%) were common to the two samples (Fig. 6e; Supplementary Data 2). The algorithm identified 17,324 and 32,651 peaks in CMML1900 and CMML1818 monocytes respectively, with 4,424 (18%) common peaks (Fig. 6f; Supplementary Data 3). Comparison of peak localization in monocytes from controls and CMML with $trTET2^{MUT}$ at high VAF identified only 29 common peaks (Fig. 6g; Supplementary Data 4), whereas comparison of peak localization in healthy donors and CMML1268 monocytes identified 1,510 common peaks (Fig. 6h; Supplementary Data 5), further arguing for CMML1268 being closer than other CMMLs to TET2 wild-type monocytes regarding EGR1 peak localization (Fig. 6e). Gene Ontology (GO) analysis of EGR1-interacting genes, using peaks that are common to healthy donor and CMML monocytes, showed a dramatic change in CMML1900 and CMML1818, with a global enrichment in genes involved in RNA processing, ribosome biogenesis, and translation (Supplementary Fig. 8). Together, these experiments indicate that $trTET2^{MUT}$ mutants with decreased expression of *TET2* gene induce chromatin remodeling that, at the level of *MIF* gene TSS, promotes the recruitment of EGR1 that may account for its overproduction.

**MIF favors monocyte differentiation of myeloid progenitors.** *Tet2* deletion was associated with an increased monocyte count in ageing mice[20]. We analyzed transcriptomic data generated from CD34$^+$ cells sorted from healthy donor and CMML patients with and without *TET2* mutation (Supplementary Table 4 and

Supplementary Data 6). Interestingly, SPI1/PU.1 transcription factor expression, promoting monocytic differentiation, was increased, while the expression of GFI1, involved in granulocytic differentiation, was decreased in $TET2^{MUT}$ compared to $TET2^{WT}$ CD34$^+$ cells (Fig. 7a). An inverse correlation was observed for the expression of these two genes in CD34$^+$ samples (Fig. 7b). These observations could account for the increase monocytes to granulocytes ratio measured in the peripheral blood of $TET2^{MUT}$ CMML patients (Fig. 7c). Since mutations decreasing *TET2* gene expression increased MIF production, and MIF could increase SPI1/PU.1 dependent transcriptional activity[52], we explored whether MIF affects myeloid cell differentiation. Cord blood CD34$^+$ cells were cultured with SCF, FLT3L, IL-3, and G-CSF in the presence or absence of 20 ng/mL recombinant MIF for 48 h before bulk RNA-seq analysis: 33 genes were differentially expressed in the presence of MIF (Supplementary Fig. 9a). GO Molecular Function analysis of these differentially deregulated genes (DEGs) identified kinase activity ($p$ value = $3.44e^{-6}$; FDR $q$ value = $2.94e^{-3}$) (Supplementary Fig. 9b) and signaling receptor binding ($p$ value = $3.34e^{-5}$; FDR $q$ value = $1.43e^{-2}$) (Supplementary Fig. 9c) signatures. GO Biological Process analysis of these DEGs identified cell morphogenesis involved in differentiation ($p$ value = $3.33e^{-6}$; FDR $q$ value = $6.23e^{-3}$) (Supplementary Fig. 9d) and anatomical structure formation involved in morphogenesis ($p$ value = $4.43e^{-5}$; FDR $q$ value = $2.76e^{-2}$) (Supplementary Fig. 9e) signatures. Finally, CIBERSORT analysis identified more monocytes and macrophages in the culture when MIF was added (Supplementary Fig. 9f), further suggesting that MIF may promote monocytic differentiation of stem and progenitor cells. Cord blood CD34$^+$ cells were cultured in medium with SCF, FLT3L, IL-3, and G-CSF in the presence or absence of recombinant MIF for 6 to 9 days before flow cytometry analysis. The addition of recombinant MIF significantly increased the fraction of CD14$^+$ monocyte cells, at the expense of CD15$^+$ granulocytes (Fig. 7d). Single cell analysis of cells collected at day 7 showed 7 clusters defined by the expression of characteristic genes (Supplementary Fig. 10a) and further validated by the ten most expressed genes (Supplementary Fig. 10b). MIF induced an increase in the number of cells in cluster 6 (monocytes) while this number was decreased in cluster 4 (granulocytes; Fig. 7e). Accordingly, MIF increased the expression in *CSF1R* gene, a SPI1/PU.1 transcription factor target, in cluster 6 while decreasing the expression of *GFI1* gene in cluster 4 (Fig. 7f). These results further argue for the ability of MIF to promote the expansion of monocytes at the expense of granulocytes (Fig. 7g). Altogether, these data suggested that MIF overproduced by *TET2*-mutant cells could promote monocyte differentiation, generating a feedback loop that may promote disease progression.

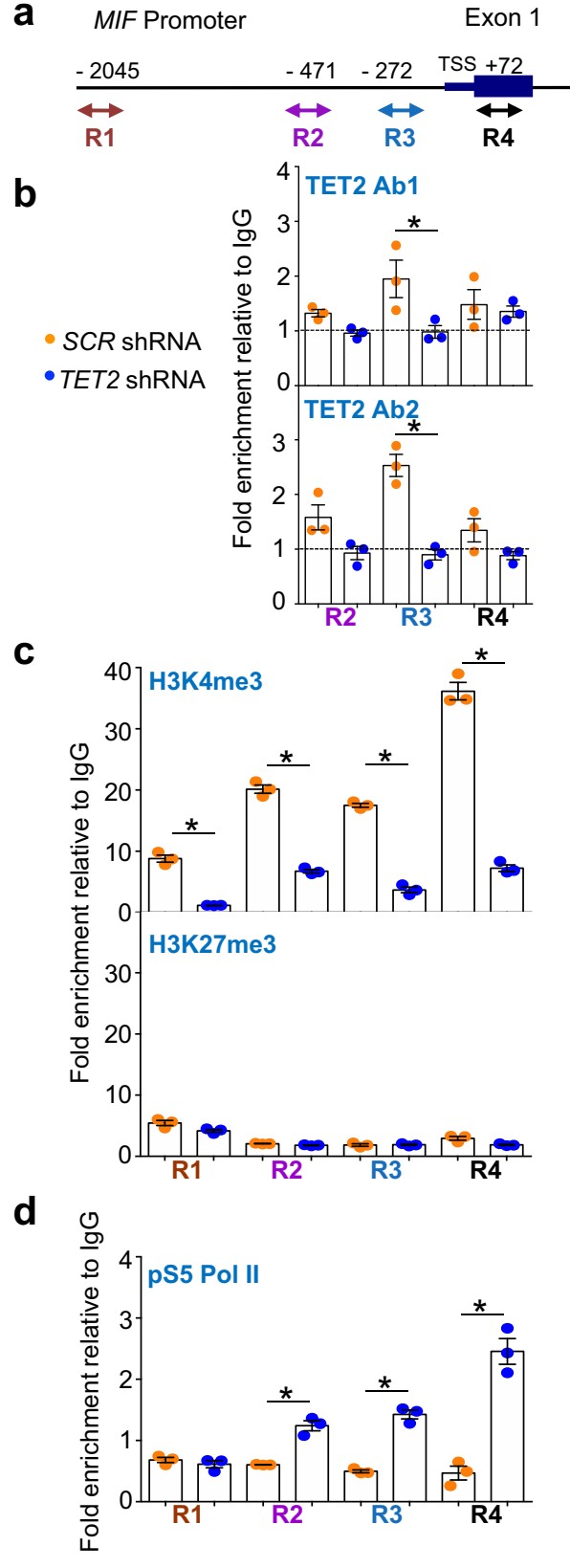

**Fig. 3 TET2 binds to the proximal promoter of *MIF*. a** Schematic representation of *MIF* gene promoter. Quantitative ChIP-PCR analyses were performed in kasumi-1 cells infected by *SCR* (yellow) or *TET2* (blue) shRNAs by using a set of primers targeting regions located in distal (R1), intermediate (R2), proximal (R3) promoter, and in exon 1, closed to the TSS (R4). **b** TET2-associated DNA was immunoprecipitated with either C2 (TET2 Ab1 upper panel) or sc-136926 (TET2 Ab2, lower panel) antibodies. **c** H3K4me3 (upper panel) or H3K27me3 (lower panel) marks associated with DNA immunoprecipitation. **d** RNA polymerase II phosphorylated on CTD serine 5 (pS5 Pol II) associated to DNA immunoprecipitation. **b–d** Results are expressed as fold enrichment relative to DNA immunoprecipitated with control immunoglobin G (IgG). Data are mean +/− SEM of each sample run in triplicate. Unpaired t Test, *$P < 0.05$.

cells into monocytes. Given the role of monocytes and MIF in atherosclerosis pathogenesis[41], MIF overproduction might also account for the cardiovascular risk identified in patients with *TET2* mutated clonal hematopoiesis[29].

TET2 diversely modulates the expression of cytokine genes, according to the cell type and their environment. In immune cells such as dendritic cells and macrophages[13,53] as well as in microglia cells[54], Toll-like receptor (TLR) agonists upregulate *TET2* expression. Conversely, TET2 binding to target genes, such as *Il6*, downregulates their expression in a catalytic activity-independent manner. TET2 favors inflammation resolution acting on immune cells and downregulating IL-6 production through HDAC1 and HDAC2 recruitment to *IL6* gene promoter[13,53]. TET2 also stimulates the inflammatory response of microglial cells[54]. In a hypoxic tumor microenvironment, upregulated TET2 promotes *IL6* gene expression in tumor cells, an effect that may involve the catalytical activity of dioxygenase since it is associated with the demethylation of *IL6* promoter[55]. Here, we show that, in the absence of inflammatory cue, *TET2* down-regulation in myeloid cells promotes the displacement HDAC1/HDAC2 from *MIF* promoter, leading to *MIF* gene expression. Strengthening this observation, HDAC inhibitors strongly decreased MIF expression in a variety of malignant cell lines and primary cells[56]. Importantly, TET2 regulates *MIF* gene expression in resting monocytes while *TET2*-mediated *IL6* regulation was detected at the recovery phase of macrophage activation[13]. MIF is the main cytokine overproduced by resting monocytes in which TET2 expression is decreased, suggesting that overproduction of other cytokines by *TET2*-mutated cells may involve less direct effects of *TET2* mutation.

Unlike TET1 and TET3, TET2 does not have a canonical CXXC domain binding unmethylated CpG, and *IDAX/CXXC4* gene, leading to TET2 interaction with CpG islands[57], is not detectable in human monocytes (our data). Alternative candidates include the NF-κB inhibitor zeta (IκBζ) that targets Tet2 to *Il6* gene proximal promoter in mouse myeloid cells exposed to lipopolysaccharides[13], the transcription factor forkhead box O3 (Foxo3a) that interacts with Tet2 to promote the proliferation of mouse neural stem cells[58], and recently, the transcription factor EGR2, another member of EGR transcription factor family, that acts as an epigenetic pioneer to recruit TET2 to its binding sites in IL4/GM-CSF-differentiated dendritic cells from primary human monocytes[7]. Here, we show that EGR1 transcription factor recruits TET2 at the *MIF* promoter in human primary monocytes.

Current evidence from animal models and clinical observations indicate that *TET2* inactivation in hematopoietic stem cells may be an early event in the initiation of myeloid malignancies, and that additional hits are necessary for tumor progression[18]. Inflammatory signals may provide these hits, as suggested in a

## Discussion

The present study identifies a role for TET2 protein in the transcriptional regulation of *MIF* gene thanks to EGR1 transcription factor in monocytes. *TET2* deficiency, e.g., as a consequence of truncating mutation, promotes *MIF* gene expression. In turn, the secreted cytokine favors the differentiation of CD34+

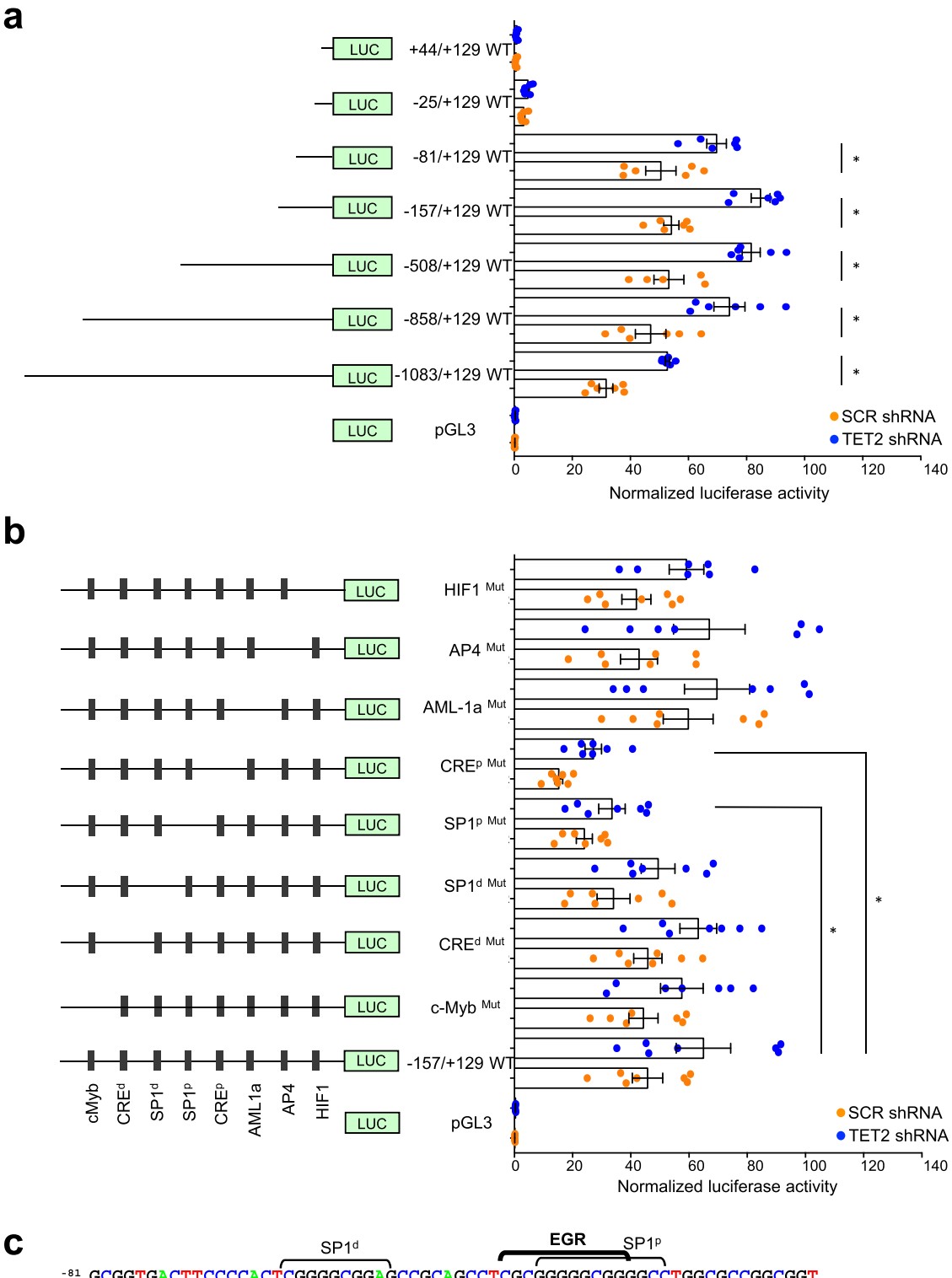

**Fig. 4 Identification of a minimal proximal promoter of *MIF* gene. a** Kasumi-1 cells stably transduced with *SCR* and *TET2* shRNAs were transiently transfected with the pGL3 vector encoding the *Luciferase* gene alone (LUC) or *LUC* gene under control of indicated *MIF* promoter fragments. Cells were co-transfected with a Renilla luciferase construct for normalization and results are the ratio between luciferase and Renilla activities. Data are mean $+/-$ SEM from at least six independent well experiments; Unpaired Students-t test, *$P < 0.05$. **b** The same experiments were performed using the $-157/+129$ MIF promoter sequence in which mutations were induced in the DNA binding sites c-Myb, CRE$^d$, CRE$^p$, SP1$^d$, SP1$^p$, AML1a, AP4, and HIF1. Data are mean $+/-$ SEM from seven independent well experiments; Unpaired t test, *$P < 0.05$. **c** DNA consensus sequences for SP1$^d$, SP1$^p$, and EGR transcription factors in *MIF* proximal promoter.

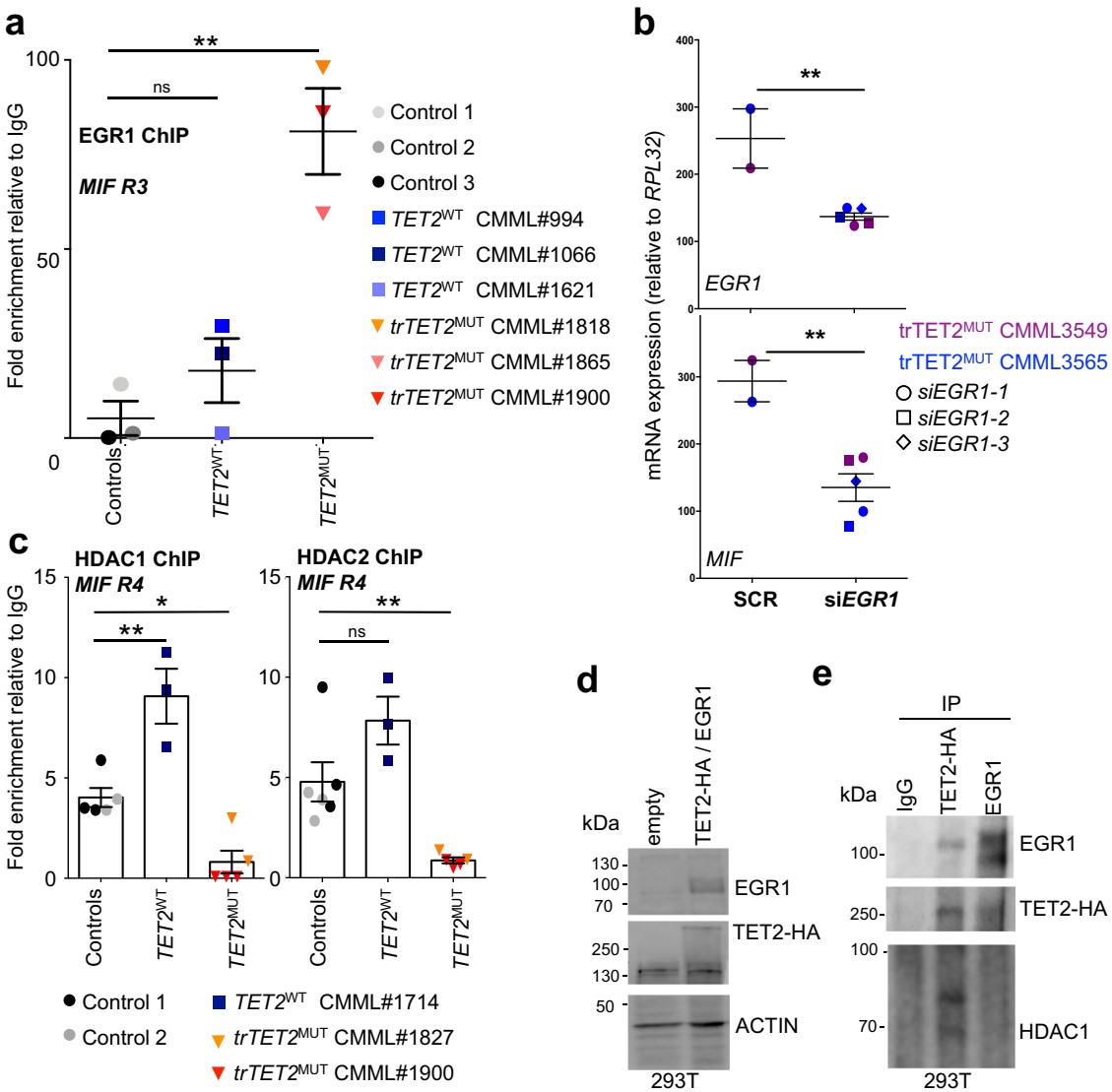

**Fig. 5 EGR1 is recruited to the MIF promoter in trTET2-mutant CMML monocytes. a** Quantitative ChIP-PCR analyses of *MIF* promoter (R3) were performed in sorted peripheral blood monocytes collected from three healthy donors (controls, circles) and 6 CMML patients, including 3 *TET2*[WT] (squares) and 3 *trTET2*[MUT] (triangles) cases. EGR1-associated DNA was immunoprecipitated. Results are expressed as fold enrichment relative to DNA immunoprecipitated with control immunoglobin G (IgG). Data are mean +/− SEM of each sample run in triplicate. Dunnett's multiple comparison test using control monocytes as control, ns, non-significant, **$P < 0.01$. **b** siRNA-mediated downregulation of EGR1 in *trTET2*[MUT] monocytes isolated from CMML3549 (purple) and CMML3565 (blue) with 2 and 3 sets of *EGR1*-siRNA respectively. RT-qPCR analysis of *EGR1* and *MIF* mRNA expression normalized to *RPL32*. Data are mean +/− SEM of each sample run in triplicate. Unpaired Students-t test using control monocytes transfected with SCR as control, **$P < 0.01$. **c** Quantitative ChIP-PCR analyses of *MIF* promoter performed in 2 controls, one *TET2*[WT] CMML and 2 *trTET2*[MUT] CMML using antibodies against HDAC1 and HDAC2, focusing on R4 of *MIF* gene. Data are mean +/− SEM of each sample run in duplicate or triplicate. Dunnett's multiple comparison test using control monocytes as control, ns non-significant, *$P < 0.05$, **$P < 0.01$. **d** 293T cells were transfected using Lipofectamine 2000 with empty vectors or pcDNA3-EGR1 and pcDNA3-HA-TET2. Immunoblot analysis of EGR1 and HA expression in transfected 293T cell. Actin, loading control. **e** Co-immunoprecipitation experiments in 293T-transfected cells. An anti-HA (HA 16B12, Covance), an anti-EGR1 (303–390 A, Bethyl Laboratories) or a control IgG were used for IP, followed by immunoblotting with anti-EGR1, anti-HA, and HDAC1 (#39531, Active Motif).

mouse model of human chronic myelogenous leukemia in which IL-6 secreted by mature myeloid cells contributes to leukemic progenitor cell development[59,60]. *Tet2* is upregulated by innate immune cells challenged with LPS and its knockdown delays inflammation resolution by precluding the repression of inflammatory cytokines as demonstrated for IL-6[13,33]. In the context of HIV-1 infection, TET2 protein degradation via Vpr protein similarly sustains IL-6 production, which enhances virus replication[53]. We noticed that, in the absence of additional infectious or inflammatory stimulus, MIF is the main cytokine overproduced by *Tet2*-deleted mice, as well as *TET2*

downregulated leukemic cells and *TET2*-mutated CMML monocytes. Many parameters may influence the level of *TET2* gene expression. In CMML patients with a *TET2* variant, truncation of the protein and the VAF of the mutated allele play essential roles. *TET2* gene expression can also be decreased in the absence of gene mutation through poorly understood mechanisms. The multiple effects of MIF on immune cells have been largely investigated[61]. However, MIF effects on hematopoietic stem and progenitor cells remain poorly understood. In a mouse retroviral model of AML, MIF produced by a *FLT3*-mutated subclone was observed to favor the expansion of leukemia-

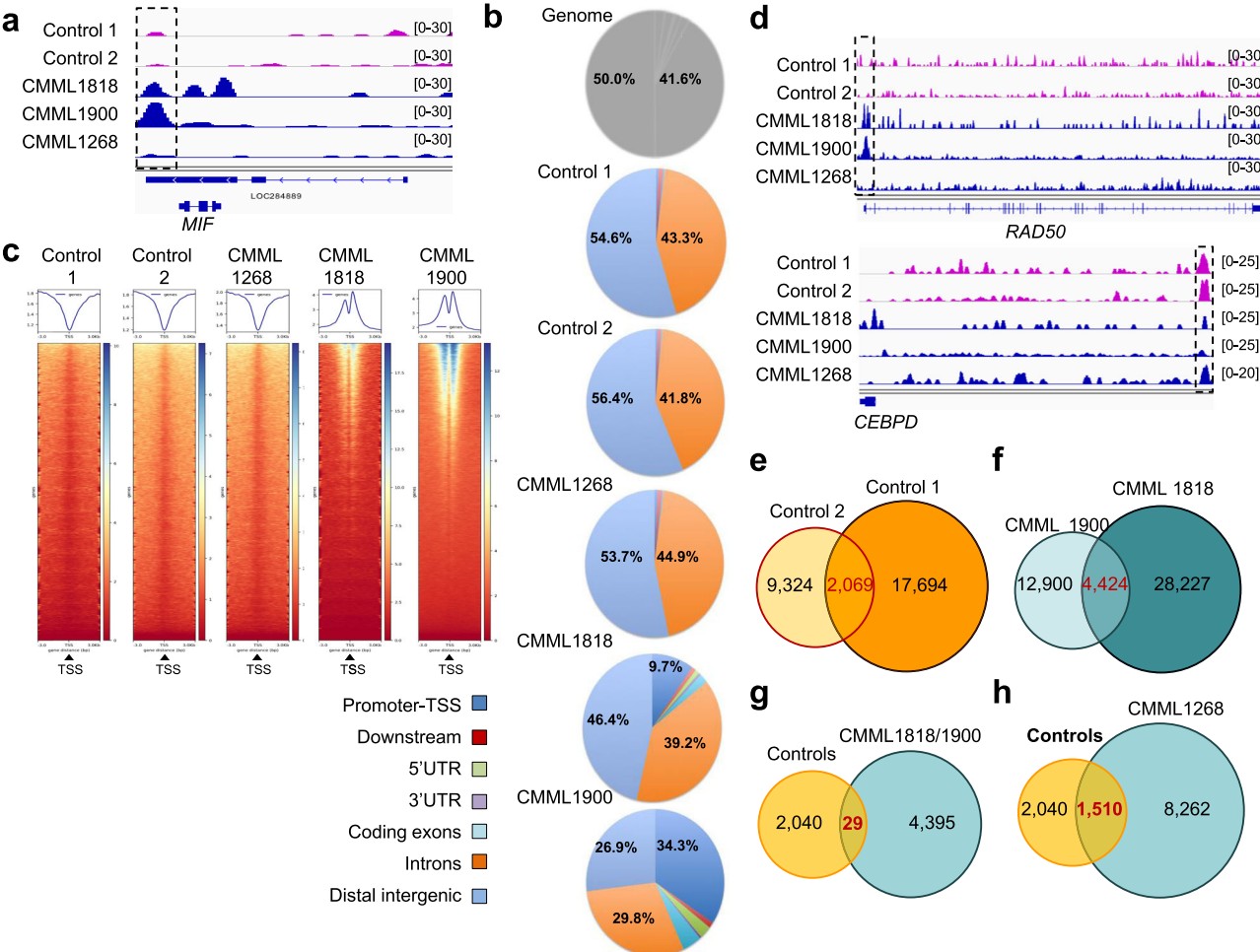

**Fig. 6 EGR1 recruitment to DNA is modified by *TET2* mutation in monocytes.** ChIP-seq experiments were performed using an anti-EGR1 antibody in sorted peripheral blood monocytes from 2 healthy donors (control 1 and 2) and three CMML patients (CMML1268: *TET2* H800fs, VAF = 16%; CMML1818: *TET2* A1241fsX1, VAF = 40% and *TET2* E1513GfsX9, VAF = 49%; CMML1900: *TET2* W564X, VAF = 100%). **a** Peak calling for EGR1 on *MIF* gene in controls (pink) and CMML (dark blue). **b** Repartition of EGR1 peaks on the genome compared to hg19 reference annotation (genome). **c** Ranking heatmaps of EGR1 peaks centered on gene transcription starting sites (TSS). **d** Peak calling for EGR1 in control (pink) and CMML (dark blue) monocytes on *RAD50* and *CEBPD* genes. **e** Venn diagram of peak calling in two healthy donors and **f** two CMML samples with high VAF tr*TET2^{MUT}*. **g** Venn diagram of peak calling comparing two controls and two CMML samples with high VAF tr*TET2^{MUT}* or **h** with the CMML sample with low VAF tr*TET2^{MUT}*.

initiating cells[62]. Here, we show that, when added to human healthy CD34[+] cells in culture, MIF promotes their differentiation into monocytes, suggesting an autocrine/paracrine feedback loop in which MIF produced by mature clonal cells creates a microenvironment that promotes the development of a myeloproliferative syndrome.

MIF levels rise during infectious, inflammatory, and autoimmune diseases. MIF promotes carcinogenesis and plays a central role in atherosclerosis pathogenesis[41,63]. The atherogenic process is initiated by endothelial dysfunction followed by an accumulation of oxidized low-density lipoproteins and an inflammatory cell infiltrate in which monocytes dominate[64]. In mouse models, *Tet2* deletion accelerates atherosclerosis through enhanced secretion of cytokines, i.e., Il-8 and Il-1β[30,65] whereas *Mif* deletion reduces the aortic inflammatory response[44]. The overexpression of MIF in *TET2*-mutated monocytes, identified in the present study, could both promote the recruitment of additional inflammatory cells in atherosclerotic lesions and favor the monocyte production of IL-1β in response to inflammatory stimuli through its direct role in NRLP3 inflammasome activation[47,61]. Of note, a G/C single-nucleotide polymorphism

(rs755622) at position −173 of human *MIF* gene is associated to a higher susceptibility to develop coronary diseases and cancers[43,66]. It would be worth determining whether this polymorphism impacts the outcome of *TET2*-mutated CHIP, including the development of coronary diseases and overt myeloid malignancies.

Together, we have shown that *TET2*-truncating mutations allows *MIF* gene expression through a catalytic domain-independent mechanism and provoke a chronic MIF overproduction by peripheral blood monocytes, in the absence of inflammatory signals. MIF could potentially promote the development of a myeloproliferative syndrome while it may accelerate atherosclerosis development. Anti-MIF antibodies were shown to exert protective effects in models of sepsis[67]. Imalumab (BAX69), one of these humanized anti-MIF monoclonal antibodies, has completed phase I and II clinical testing with acceptable toxicity[68]. As TET2 by itself is hardly druggable, targeting MIF should be considered as an alternative therapeutic strategy for preventing the development of atherosclerotic lesions and chronic myeloid malignancies in individuals with *TET2*-truncating mutations.

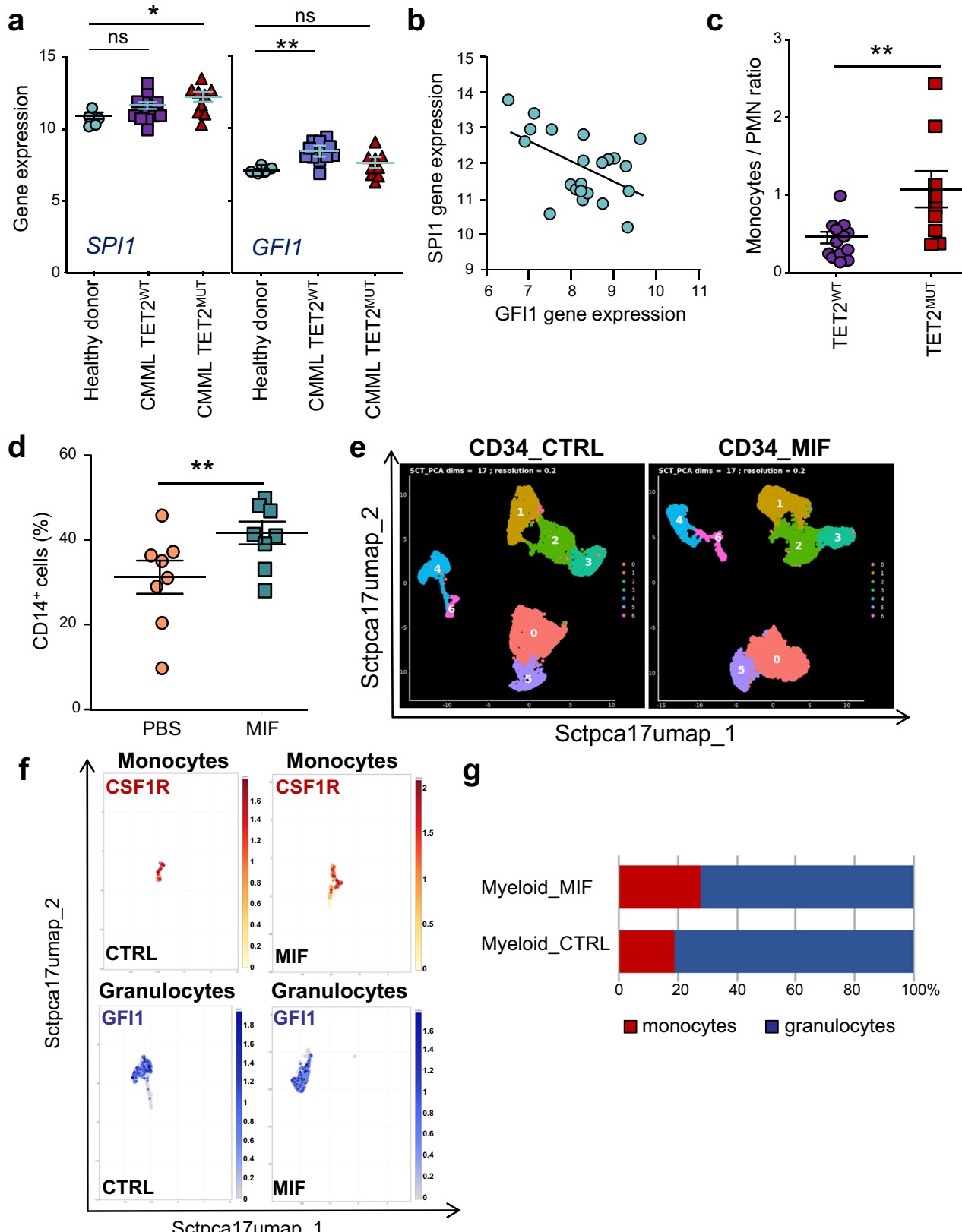

## Methods

**In vitro granulo-monocyte differentiation**. Umbilical cord blood samples were collected from healthy newborns with mother consent: AP-HP, Hôpital Saint-Louis, Unité de Thérapie Cellulaire, CRB-Banque de Sang de Cordon, Paris, France – N° d'autorisation: AC-2016-2759. Sorted CD34$^+$ cells, using magnetic beads and AutoMacs system (Miltenyi Biotech), were depleted for *TET2* and subjected to monocyte/macrophage differentiation as previously described[48]. Supernatants were

collected at indicated day. For Fig. 7d, sorted CD34$^+$ were cultured 24 h at $1 \times 10^6$ cells/mL in MEM-alpha medium (Thermo Fisher Scientific) with 10% heat inactivated fetal bovine serum (FBS), 1% penicillin/streptomycin and 2 mM L-Glutamine (Thermo Fisher Scientific), stem cell factor recombinant (SCF, 50 ng/mL; Immunex), interleukin-3 recombinant (IL-3, 10 ng/mL; Novartis), IL-6 recombinant (10 ng/mL; Peprotech), thrombopoietin recombinant (TPO, 10 ng/mL; Peprotech), Fms-like tyrosine kinase 3 recombinant (FLT-3, 50 ng/mL;

**Fig. 7 MIF promotes monocyte differentiation. a** CD34$^+$ cells were sorted from the peripheral blood of healthy donor ($n = 5$) and CMML patients, without ($n = 13$) or with TET2 ($n = 9$) mutation, and analyzed by Agilent microarrays. Indicated gene expression was monitored (Log2 intensity). Data are mean $+/-$ SEM of indicated biological samples. Dunnett's multiple comparison test using healthy donor as control, *$P < 0.05$, ns, non-significant. **b** Inverse correlation between *SPI1* and *GFI1* Log2 intensity expression in CD34$^+$ samples. $R^2 = 0.24$; $p = 0.02$. **c** Monocytes to polymorphonuclear cell ratio in the peripheral blood of CMML patients without ($n = 13$) or with TET2 ($n = 9$) mutation. Mann–Whitney test: **$P < 0.005$. **d** Cord blood CD34$^+$ cells were induced to differentiate as described in Fig. 1, in the presence or the absence of 20 ng/mL MIF for 5–9 days before measuring the fraction of CD14$^+$ cells by flow cytometry. Data are mean $+/-$ SEM from eight independent experiments; Mann–Whitney test: **$P < 0.005$. **e** Single cell analysis of cord blood CD34$^+$ cells induced to differentiate as described in Fig. 1 with or without MIF at day 7. Umaps of cluster analysis at 17 dimensions with a resolution of 0.2. **f** Umaps of *CSF1R* and *GFI1* gene expressions in monocyte and granulocyte clusters in CTRL and MIF conditions. Scales indicate the intensity of expression. **g** Percentage of monocytes and granulocytes in the myeloid compartment in CTRL and MIF conditions.

---

Diaclone), granulocyte colony-stimulating factor recombinant (G-CSF 10 ng/mL; Peprotech), in a 37 °C incubator with 5% CO$_2$. MIF (289-MF-002, R&D systems; 20 ng/mL every two days) was added to medium for 5–9 days before immuno-phenotypic analysis (Supplementary Table 5 for antibodies) using a Fortessa flow cytometer (BD biosciences). Analysis was performed using Kaluza Software (Beckman Coulter) with gating strategy indicated in (Supplementary Fig. 11).

**Cytokine profiler array and MIF immunoassay.** CD34$^+$ collected supernatants were analyzed using human cytokine antibody array (panel A; R&D Systems). MIF concentrations were quantified by ELISA (human: MIF Quantikine ELISA Kit, R&D Systems; mouse: Mif USCNK life Science). Supernatants from fresh bone marrow samples were frozen at −80 °C until MIF analysis using mesoscale technology (Meso Scale Diagnostics). CD14$^+$ monocyte cells were plated at 500,000 cells/ml in serum free RPMI medium during 18 h and collected supernatants were stored at −80 °C until analysis with human cytokine antibody array (panel A; R&D Systems). Serum samples from age-matched controls and TET2-mutated CMML patients were collected and stored at −80 °C until analysis with human cytokine antibody array (panel A; R&D Systems).

**Cell culture and derivation of TET2 deficient cell lines.** Kasumi-1 (CRL-2724), M07e (CRL-7442), TF-1 (CRL-2003) cell lines (American Type Culture Collection), and UT-7 (kindly provided by Dr. Patrick Mayeux, Cochin Hospital, Paris, France), which all express wildtype *TET2*, were maintained in RPMI1640 or MEMα medium (Gibco) with 10 mM L-Glutamine, 10 mM penicillin-streptomycin and 20% (v/v) FBS (Hyclone) with or without GM-CSF (100 ng/ml, Peprotech). Modified cell lines with shRNA-TET2 (5′-GGGTAAGCCAAGAAAGAAA-3′) or shRNA-scramble (SCR) (5′-GCCGGCAGCTAGCGACGCCAT-3′) were previously described[48].

**Immunoblotting.** Cells were lysed for 15 min at 4 °C in lysis buffer (150 mM NaCl, 50 mM TRIS pH 7.8, 1% NP40, 1 mM EDTA, 1 mM PMSF, NAF, orthovanadate, Protease Inhibitor Complete (Roche). 15 µg of total proteins were separated on polyacrylamide gel and transferred to nitrocellulose membrane (ThermoFisher Scientific). Membranes were blocked with 5% bovine serum albumin in PBS, with 0.1% Tween-20 (Sigma-Aldrich) for 40 min at RT, incubated overnight at 4 °C with the primary antibodies (dilution 1/1000e), washed in PBS-0.1% Tween-20, incubated further with HRP-conjugated secondary antibody (400 ng/mL) for 1 h at RT and washed again before analysis using Immobilon Western Chemiluminescent HPR Substrate system (Millipore). Actin (A5441, Sigma-Aldrich) and MIF (sc-80191, Santa Cruz Biotechnology) antibodies were used for detection and quantification were performed using Image J software (http://imagej.nih.gov/ij).

**RT-qPCR analysis.** Total RNA was extracted with RNeasy Mini Kit (Qiagen) and reverse transcribed with SuperScript II reverse transcriptase (Thermo Fisher Scientific) with random hexamers (Thermo Fisher Scientific). Real-time quantitative polymerase chain reaction (RT-qPCR) was performed with AmpliTaq Gold polymerase in an Applied Biosystems 7500 thermocycler using the standard SyBR Green detection protocol as outlined by the manufacturer (Thermo Fisher Scientific). Briefly, 12 ng of total cDNA, 50 nM (each) primers, and 1× SyBR Green mixture were used in a total volume of 20 µL. Primers are listed in Supplementary Table 5.

**Mice.** Animal experiments were conducted according to Gustave Roussy guidelines and authorized by the Direction Départementale des Services Vétérinaires du Val-de-Marne. Animal experiments were performed in accordance with 2010/63/UE European legislation and decree n°2013-118 of French legislation and recorded under protocol number APAFIS# 2012-018-16-540 and 2016-104-7171. Mouse models were previously described[20]. Blood collection from young mice (1–3 months) was performed under anesthesia with isofluorane at Gustave Roussy animal core facility. Complete blood counts were obtained using MS9-5 hematology analyzer (Melet Schloesing Technologies). Bone marrow cells were flushed and centrifuged at 1500 rpm for 10 min at RT before collecting supernatant that was frozen at −80 °C.

**Control and patient sample collection.** Peripheral blood samples were collected from healthy donors (Buffy coat, Etablissement Français du Sang, Rungis, France), age-matched donors, and CMML patients with informed consents in compliance with the ethical committee Ile-de-France (MYELOMONO cohort, DC-2014-2091). CMML patients were diagnosed according to the latest 2016 WHO criteria[69]. CD14$^+$ monocytes were isolated as previously described and all patient monocyte DNA included in the study were subjected to next generation sequencing (NGS) for a myeloid panel, by previously described methods[50]. CMML patient clinico-biological characteristics, summarized in Tables S2–S4 and S9, were obtained from our CMML database with clinical/biological annotations (DR-2016-256) and NGS analysis of a previously described panel[19].

**RNA sequencing and analysis.** 10 age-matched healthy CD14$^+$/CD16$^-$ monocyte and 60 CMML CD14$^+$ monocyte RNAs were used to prepare libraries with Illumina TruSeq RNA Sample Prep Kit v2. All controls and CMML patients were subjected to NGS myeloid panel sequencing[50]. RNA-seq libraries were sequenced on a 75 bp pair-end run using NextSeq500 High Output Kit 150 cycles; 400 M flow cell. All transcripts from Gencode V24 were quantified for each library in transcript per million (TPM) units using the Kallisto software. Gene-level expression was then calculated as the sum of TPM values for each gene transcript. Heatmap was generated using Morpheus (https://software.broadinstitute.org/morpheus/). For cord blood CD34$^+$ cell RNA-seq differentiated in presence or not of MIF, the RNA integrity (RNA Integrity Score ≥7.0) was checked on the Agilent Fragment Analyzer (Agilent) and quantity was determined using Qubit (Invitrogen). SureSelect Automated Strand Specific RNA Library Preparation Kit was used according to the manufacturer's instructions with the Bravo Platform. Briefly, 50–100 ng of total RNA sample was used for poly-A mRNA selection using oligo(dT) beads and subjected to thermal mRNA fragmentation. The fragmented mRNA samples were subjected to cDNA synthesis and were further converted into double stranded DNA using the reagents supplied in the kit, and the resulting dsDNA was used for library preparation. The final libraries were bar-coded, purified, pooled together in equal concentrations, and subjected to paired-end sequencing on Novaseq-6000 sequencer (Illumina) at Gustave Roussy. We used CIBERSORT, an analytical tool from the Alizadeh Laboratory developed by Newman et al. to provide an estimation of the abundances of member cell types in a mixed cell population, using gene expression data[70].

**Chromatin immunoprecipitation (ChIP) and ChIP-seq.** ChIP experiments were performed using ChIP-IT kit (Active Motif) as previously described[71]. 5 µg of antibodies (Supplementary Table 5) were used. Immunoprecipitated-chromatin was eluted after several washes, reverse cross-linked and stored at −20 °C. ChIP-qPCRs were performed in the same way as RT-qPCR with 2 µL of ChIP or IgG samples instead of cDNA. Primer sequences are in Supplementary Table 5. Enriched DNA from EGR1-ChIP and input DNA fragments were used to generate libraries as previously described[71]. Fifty-cycle single-end sequencings were performed using HiSeq 2000 (Illumina). Reads were aligned using human genome hg19 with BWA (v0.7.5a), peak calling assessed using MACS 2.0, and annotation done with HOMER (v4.7.2), with a $p$ value of 0.01. Integrative Genomics Viewer (IGV 2.1) was used for representation.

**Luciferase assays.** Kasumi-1 cells were transfected with the reporter plasmid pGL3-MIF-luc and diverse fragments of *MIF* gene promoter[51], with or without mutation in transcription factor binding motifs (Supplementary Table 5) and co-transfected with TK-Renilla reporter for normalization of transfection efficiency. Using dual luciferase assay, luciferase activity was measured with a luminometer (Promega).

**Bisulfite DNA treatment and sequencing.** Genomic DNA was isolated from monocytes of controls or CMML patients using QIAGEN's standard procedures. Two hundred nanograms of total genomic DNA was modified by bisulfite treatment according to the manufacturer's instructions (MethylDetector, Active Motif). Converted *MIF* promoter was identified by PCR with converted primers forward

(5′-GGTGATTTAGTGAAAGGATTAAGAA-3′) and reverse (5′- CATAATAACA AAAAAACCAAAAAACCC-3′), and direct sequencing reaction was performed using standard conditions according to the manufacturer's instructions (Applied Biosystems).

**EGR1 knockdown in TET2-truncated mutated CMML monocytes**. We used Stealth siRNAduplex (ThermoFisher Scientific) targeting EGR1 and Stealth RNAi™ siRNA negative control (SCR) introduced into CMML monocytes by Lipofecta-mine 2000 (ThermoFisher Scientific). Briefly, $1 \times 10^6$ cells in 500 µl in a 24-well plate were transfected according to the manufacturer's recommendations with 1nmol siRNA.

siRNAs targeting EGR1:
*siEGR1-1*: 5′-UCUCCCAGGACAAUUGAAAUUUGCU-3′;
*siEGR1-2*: 5′-AGCAAAUUUCAAUUGUCCUGGGAGA-3′;
*siEGR1-3*: 5′-GAUCUCUGACCCGUUCGGAUCCUUU-3′.
Knock-down efficacy was performed using RT-qPCR and MIF expression was addressed.

**293 T transient transfection**. 293T cells were transfected with pcDNA3-HA-TET2 and pcDNA3-EGR1 plasmids using Lipofectamine 2000 according to the manufacturer's recommendation. 72 h after transfection, cells were harvested and lysed 20 min on ice in a buffer containing 50 mM Tris pH 7.4, 150 mM NaCl, 1% NP-40, 10% Glycerol, 1 mM Na3VO4, and 1× Protease inhibitor cocktail (100 µL for $10 \times 10^6$ cells). Samples were then centrifuged 15 min at 4 °C at 18,000 × g and the supernatant containing the proteins was collected. 10 µg of anti-HA or anti-EGR1 antibody or negative control IgG were added and the samples incubated one night at 4 °C with agitation. 100 µL of Protein A/G PLUS-Agarose (Santa Cruz biotechnology) were incubated with the samples during 2 h at 4 °C with agitation. The complexes were then precipitated with 1 min centrifugation at 2000 × g and the supernatant was removed. The remaining beads were washed 5 times with 500 µL of lysis buffer without NP-40. 2× Laemmli with 0.1 M DTT and Protease inhibitor cocktail 1× was added on the beads and outputs. The samples were then boiled 10 min at 95 °C. Following SDS-PAGE and blotting, the membranes were incubated with either an anti-EGR1 antibody, an anti-HA, or an anti-HDAC1 antibody.

**Immunoprecipitation experiment in primary monocytes**. CD14+ cells were lysed 20 min on ice in a buffer containing 50 mM Tris pH 7.4, 150 mM NaCl, 1% NP-40, 10% Glycerol, 1 mM Na3VO4 and 1× Protease inhibitor cocktail (100 µL for $10 \times 106$ cells). Samples were then centrifuged 15 min at 4 °C at 18,000 × g and the supernatant containing the proteins was collected. 10 µg of anti-HDAC1 (#39531, Active Motif) or an anti-HDAC2 (#39533, Active Motif) or negative control IgG were added for $10 \times 106$ cells and the samples incubated one night at 4 °C under agitation. One hundred µL of Protein A/G PLUS-Agarose (Santa Cruz Biotechnology) were incubated with the samples during 2 h at 4 °C under agitation. The beads were washed 5 times with 500 µL of lysis buffer without NP-40. 2X Laemmli with 0.1 M DTT and Protease inhibitor cocktail 1X was added on the beads and boiled 10 min at 95 °C. Proteins were separated on polyacrylamide gel and transferred to nitrocellulose membrane (ThermoFisher Scientific). Membranes were blocked with 5% bovine serum albumin in PBS, with 0.1% Tween-20 (Sigma-Aldrich) for 40 min at RT, incubated overnight at 4 °C with anti-TET2 antibody (sc-136926, Santa Cruz biotechnology) (dilution 1/1000e), washed in PBS-0.1% Tween-20, incubated further with HRP-conjugated secondary antibody (400 ng/mL) for 1 h at RT and washed again before analysis using Immobilon Western Chemiluminescent HPR Substrate system (Millipore, Molsheim, France).

**Gene expression microarray analysis**. Gene expression in purified CD34⁺ cells was analyzed with Agilent® SurePrint G3 Human GE 8x60K Microarray (Agilent Technologies, AMADID-28004). After single color hybridization and array scan-ning, microarray images were analyzed using Feature Extraction software version (10.7.3.1) with default settings. Using LIMMA R package, data were normalized by the quantile method and analyzed (for single value of each transcript, the mean of each replicated probe was taken).

**3′ Single-cell RNAseq**. Sample preparation was done at room temperature. Single-cell suspensions were loaded onto a Chromium Single Cell Chip (10x Genomics) according to the manufacturer's instructions for co-encapsulation with barcoded Gel Beads at a target capture rate of ~10,000 individual cells per sample. Captured mRNAs were barcoded during cDNA synthesis using the Chromium Next GEM Single Cell 3′ GEM, Library & Gel Bead Kit v3.1 (10X Genomics) according to the manufacturer's instructions. All samples were processed simultaneously with the Chromium Controller (10X Genomics) and the resulting libraries were prepared in parallel in a single batch. We pooled all of the libraries for sequencing in a single SP Illumina flow cell. All of the libraries were sequenced with an 8-base index read, a 28-base Read1 containing cell-identifying barcodes and unique molecular identi-fiers (UMIs), and a 91-base Read2 containing transcript sequences on an Illumina NovaSeq 6000.

**3′ Single-cell RNAseq analysis**. Raw BCL-files were demultiplexed and converted to Fastq format using bcl2fastq (version 2.20.0.422 from Illumina). Reads quality control was performed using fastqc (version 0.11.9) and assignment to the expected genome species evaluated with fastq-screen (version 0.14.0). Reads were pseudo-mapped to the Ensembl reference transcriptome v99 corresponding to the homo sapiens GRCh38 build with kallisto (version 0.46.2) using its 'bus' subcommand and parameters corresponding to the 10X Chromium 3′ scRNA-Seq v3 chemistry. The index was made with the kb-python (version 0.24.4) wrapper of kallisto. Barcode correction using whitelist provided by the manufacturer (10X Genomics) and gene-based reads quantification was performed with BUStools (version 0.40.0).

Cell barcode by symbol count table was loaded in R (version 4.0.4) using the BUSpaRse package (version 1.5.3). To call real cells from empty droplets, we used the emptyDrops function from the dropletUtils package (version 1.10.3), which assesses whether the RNA content associated with a cell barcode is significantly distinct from the ambient background RNA present within each sample. Barcodes with $p$-value < 0.001 (Benjamini–Hochberg-corrected) were considered as legitimate cells for further analysis.

The count matrix was filtered to exclude genes detected in less than five cells, cells with less than 1500 UMIs or less than 200 detected genes, as well as cells with mitochondrial transcripts proportion higher than 20%. Cell cycle scoring of each cell was performed using two methods: the CellcycleScoring function from the Seurat package (version 4.0.0), and the cyclone function from Scran (version 1.18.5). Barcodes corresponding to doublets were identified and discarded using the union of two methods: scDblFinder (version 1.4.0) using default parameters, and scds (version 1.6.0) with its hybrid method using default parameters. We manually verified that the cells identified as doublets did not systematically correspond to cells in G2M phase.

Seurat (version 4.0.0) was applied for further data processing. The SCTransform normalization method was used to normalize, scale, select 3000 Highly Variable Genes and regress out bias factors (the number of detected transcripts and the proportion of ribosomal transcripts). The number of PCA dimensions to keep for further analysis was evaluated by assessing a range of reduced PCA spaces using 3 to 49 dimensions, with a step of 2. For each generated PCA space, Louvain clustering of cells was performed using a range of values for the resolution parameter from 0.1 to 1.2 with a step of 0.1. The optimal space was manually evaluated as the one combination of kept dimensions and clustering resolution resolving the best structure (clusters homogeneity and compacity) in a Uniform Manifold Approximation and Projection space (UMAP). For the 2 samples, 17 dimensions were retained with a resolution of 0.2.

Marker genes for Louvain clusters were identified through a «one versus others» differential analysis using the Wilcoxon test through the FindAllMarkers function from Seurat, considering only genes with a minimum log fold-change of 0.5 in at least 75% of cells from one of the groups compared, and FDR-adjusted $p$-values < 0.05 (Benjaminin–Hochberg method). Gene visualization was done using Cerebro (version 1.2.2).

**Statistics and reproducibility**. All statistical analyses (Paired and unpaired t-tests, Mann–Whitney test, Dunnett's multiple comparison tests, correlation using linear regression) were performed using Graph-Pad Prism software (GraphPad Software, La Jolla, CA, USA). Values of $p < 0.05$ were considered statistically significant. The used statistical test, the $p$ value, and the number of samples are indicated in the respective Figure legends.

## Data availability

Control and CMML monocyte RNA-seq data have been deposited in the NCBI Gene Expression Omnibus database under accession number GSE165305 and GSE188624. ChIP-seq datasets are available in the Array Express database at EMBL-EBI (www.ebi.ac.uk/arrayexpress) under accession number E-MTAB_6305 and E-MTAB_11132. Cord blood bulk RNAseq and scRNAseq have been deposited in the European Genome-Phenome archive (EGA) under accession number EGAS00001005814. The data that support the findings of this study are available within the article and its Supplementary Information and Supplementary Data files. Uncropped western blots are available in (Supplementary Fig. 12). The source data underlying all graphs and charts are provided as Supplementary Data 7.

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

## Acknowledgements

This work was supported by INSERM and by the Ligue contre le cancer (labeled team to F.P.). F.D. received a PRTK-16-122 with N.D. as partner. 'Taxe d' apprentissage' program Genomic Core Facilities - TA2012 to R.I. A.I. was supported by PRTK-16-122 and then by a grant from the Ligue Nationale Contre le Cancer. We thank Bastien Job for bulk RNA-seq analysis. M.F. and M.A. are supported by SIRIC SOCRATE INCa-DGOS-Inserm_1255. Part of high-throughput sequencing was performed by the genomic platform of Gustave Roussy, which is supported by Cancéropole Ile de France and by the SIRIC SOCRATE program. The authors thank Margot Morabito and Jeffie Lafosse for collecting clinical and biological annotations on CMML patients, and physicians and patients of the Groupe Francophone des Myélodysplasies for providing samples. We thank for umbilical cord blood samples Hôpital Saint-Louis, Unité de Thérapie Cellu-laire, CRB-Banque de Sang de Cordon, Paris, France – N° d'autorisation: AC-2016-2759. The authors are grateful to Gustave Roussy Animal core facility and cytometry core facility for their helpful assistance in experiments. T.R. is supported by the Swiss National Science Foundation (SNSF grant number 310030_173123).

## Author contributions

Methodology: E.P., D.S.B., A.I., R.I., T.R., C.J., A.N., M.F., M.B., and N.D.; formal ana-lysis: M.E.F., M.A., D.G., and N.D.; funding acquisition: N.D., F.D., E.S., and F.P.; resources: B.B., O.W.B., R.I., O.A.B., W.V., F.D., and E.S.; discussion: F.P., O.A.B., W.V., F.D., E.S., and N.D.; concept, design, data analysis, supervision, and paper writing: N.D.

## Competing interests

The authors declare no competing interests.
