## [Peer Review File · Communications Biology]

Reviewers' comments:

Reviewer #1 (Remarks to the Author):

The manuscript entitled "Macrophage migration inhibitory factor is a deregulated cytokine through EGR1 in resting TET2-truncating mutated monocytes" identifies an inverse association between TET2 and the cytokine migration inhibitory factor (MIF). The authors contend that loss of Tet2 enables its binding partner EGR1 to bind to the MIF promoter and facilitate its elevation. They further propose that elevated levels of MIF may be responsible for increased monocyte differentiation observed in TET2-deficient diseases, such as chronic myelomonocytic leukemia (CMML) and clonal hematopoiesis of indeterminate potential (CHIP).

A significant strength of this study is the very careful demonstration of MIF RNA and protein elevation in both mouse and human models. The authors showed MIF elevation both in vitro and in vivo and via multiple methods, including cytokine array, qRT-PCR, and Western blotting. However, the leukemia cell lines used were mostly derived from acute megakaryocytic leukemia and erythroleukemia. As the authors emphasize the role of myeloid cells in the pathogenesis of TET2-associated disease, there are acute myeloid leukemia cell lines that may lend greater support to this proposed mechanism. The introduction of the manuscript is well written and provides relevant and compelling background information. The elevation of MIF represents a novel finding that may have valuable implications for the role of inflammation in TET2-associated diseases.

Weaknesses of the study include statistical considerations, minor grammatical and formatting errors, and data interpretation. Several of the human studies consist of very small numbers of patients. The translational impact of this study would be increased by the inclusion of more patients, especially considering the low penetrance, high latency, and significant variability inherent to TET2-associated diseases. The graphs quantifying the Western blots in Figure 1E do not have error bars. In Figure 1, it is surprising that elevated levels of IL-1 β and IL-6 were not detected in the cytokine array. Previous studies indicate that Tet2 loss leads to the elevation of IL-1 β secretion from proinflammatory macrophages in a mouse model of CHIP-associated cardiovascular disease. Furthermore, elevated levels of IL-1 and IL-6 have been detected in patients with CHIP-associated mutations in TET2. In addition, the authors identified three cytokines that were elevated in the absence of TET2, but only MIF was affected by altering Tet2 expression. This observation suggests that TET2 may modulate inflammatory factors via direct and indirect mechanisms; however, this notion was not addressed. In Figure 2C, the significance of including young and old healthy individuals is unclear; however, this information may be relevant to interpretations regarding CHIP. In addition, the authors did not specify if healthy individuals in their studies were screened for CHIP-associated mutations, including those in TET2. This information could have significant implications for the interpretation of this data. In Figure 2D, the authors conclude that the expression of cytokine-encoding genes other than MIF, including IL-1 β , are not altered in CMML patients. However, the heat map shows that the expression of a variety of other cytokine-encoding genes. Consistent with previous studies, this result supports the complexity and heterogeneity of the inflammatory response in patients carrying TET2 mutations. The lack of a clear relationship between MIF and other inflammatory signaling molecules in this experiment may actually diminish the clinical significance of MIF as it is unclear how MIF is affecting the inflammatory response. In addition, the authors demonstrate recruitment of EGR1 to genes involved in ribosome biogenesis, translation, and RNA processing, suggesting that inflammation may not be the key process affected by EGR1 and MIF. Furthermore, they do not address the consequences of elevated MIF signaling. Based on the functions of MIF and the proinflammatory environment in Tet2-deficient cells, it would be expected that elevation of MIF would enhance or contribute to the proinflammatory environment promoted by Tet2 loss; however, the authors do not demonstrate or address this possibility. Additional experiments, clarification of this mechanism, and discussion of these notions would enhance the impact of this study. The results in Figure 7 are not as convincing, showing a weak correlation and not demonstrating causation. However, the interpretation of these results is vital to the overall hypothesis that the elevation of MIF in the absence of TET2 drives enhanced myeloid differentiation. This data could be considerably strengthened by additional experiments. In general, the discussion section is less relevant to the current study and could be restructured. The authors consistently emphasize a potential role for MIF in cardiovascular disease. While compelling, this connection was not addressed in the current study.

The innovation of this study is the identification of a novel cytokine that is associated with TET2

mutations. While previous studies indicate a role for TET2 in modulating inflammation, MIF has not been detected, and these studies often employed bacterial challenge experiments. The significance of this study is that this new mechanism for TET2-associated disease may represent a novel therapeutic target for patients with these mutations. A unique aspect of this study is the detection of elevated MIF under resting or unchallenged conditions, providing support for an intrinsic role for TET2 in the inflammatory response. Collectively, this manuscript presents interesting novel findings that may have significant translational impact. It is recommended that the authors address the experimental concerns about the manuscript, clarify the mechanism and data interpretation, and correct grammar, typos, and formatting.

Reviewer #2 (Remarks to the Author):

Inactivating TET2 mutations are among the most common drivers of clonal hematopoiesis and myeloid neoplasms, especially chronic myelomonocytic leukemia (CMML). To this point, the inflammatory sequelae of TET2 mutations (especially regarding the increased risk of cardiovascular disease) have been mainly related to increased NLRP3 inflammasome activation and IL-1 β and IL-6 production by TET2-mutant monocytes/macrophages. In this interesting and timely manuscript, Dr. Pronier et al. use a host of complementary experiments in murine and human cells to demonstrate increased, EGR1-driven production of macrophage migration inhibitor factor (MIF) in TET2-deficient cells, potentially contributing to skewed monocytic differentiation. Although not explored in this manuscript, targeting MIF to prevent TET2 mutation-driven leukemogenesis and cardiovascular disease may warrant further study.

The following comments are intended to improve the quality of the manuscript.

1. Introduction - line 6 - "activity of TET enzyme activity" - redundant.
2. Intro - use of "ARCH" is acceptable but the risk of cardiovascular disease has been established for clonal hematopoiesis with variant allele fraction of at least 2% (i.e. definition of "CHIP"). Please specify or use "CHIP".
3. Results - paragraph 1 - "increased in four human leukemic cell lines (kasumi-1, M07e, UT-7 and TF-1) in which TET2 gene expression was decreased" - please specify "using shRNA" or something to this effect. Otherwise, the impression may be given in the text that these cell lines naturally have decreased TET2 expression. On this note, it might be helpful to clarify if these cell lines contain any known TET2 mutations.
4. Results - paragraph 1 - "excluding differential monocytes counts as a confounding factor" - please specify if differential counts were only assessed in blood, or also in bone marrow.
5. Results - p. 6 & Suppl. Fig. 2A/B - please explain why n=50 total TET2-mut in Suppl. Fig 2A but n = 38+19 in 2B.
6. Results - p. 7 and Fig. 3c - can the authors please clarify - it seems counter-intuitive that decrease of H3K4me3 (normally associated with transcriptional activation) in cells treated with TET2 shRNA would be associated with increased MIF promoter activity and transcription. Am I misunderstanding?
7. Results - p. 8, Fig. 6 & Suppl Figs. 6 - for the examples of EGR1 distribution around TSS for RAD50 and other genes listed, presumably these are relevant to the subsequent Gene Ontology analysis? If so, it may strengthen their selection to clarify. Can other factors (i.e. non-TET2 mutations) in CMML affect EGR1 peak localization?
8. Results - p. 9 - why was MIF expression not compared in transcriptomic data of CD34 cells from healthy donor and CMML patients (+/- TET2 mutations)? Is it not expressed in CD34+ cells? Was it not covered in the array?

9. Discussion - p. 11 & elsewhere - re: "in the absence of infectious or inflammatory stimulus" - can the residual ex vivo effects of possible in vivo exposure to such a stimulus be completely excluded in primary patient monocytes? Can epigenetic effects of endogenous exposures be excluded?
10. Discussion - p. 12 - what is the significance of amino acids 86-102 of MIF? It may be helpful to describe the structure/function of MIF in the Introduction or the relevance of this region in the Discussion or to exclude this detail.
11. Methods - p. 13 - what is the source of SCF, IL-3, IL-6, TPO-FLT3? i.e. vendor, species, recombinant?
12. Fig. 6 title is followed by "(a,f)". What does this indicate?
13. Fig. 1a - numbering of columns stops at 12, when 20 are present.
14. Am I missing the data proving that shTET2 knocks down TET2 expression at mRNA and/or protein level?
15. Fig. 2 - can the authors discuss why some healthy donors and TET2-WT CMML patients have high expression of MIF? And why do TET2-MUT CMML patients not uniformly over-express MIF? This doesn't seem to be fully explained by truncating vs non-truncating mutations. What about confounding pathways? Is it possible other mutations (non-TET2) could increase MIF or interfere with tr-TET2-mutations?
16. Fig. 2D - a selection of cytokines & chemokines are shown. Are the full list of targets described somewhere in the manuscript?
17. Fig. 2G - it is understood why bone marrow plasma is shown but have the authors also compared MIF expression in blood plasma or serum?
18. Fig. 5 a and b - were insufficient cells available to run CMML samples in parallel in parts a and b? Only CMML 1900 overlaps in both panels.
19. Fig.7 & related manuscript section - the authors cite a reference that MIF can increase SPI1/PU.1 but don't close the loop here (showing either increased SPI1 or decreased GFI1 in this culture system). Similarly, they talk about a feedback loop but don't try to interfere with MIF (or EGR1) in their experiments (i.e. CRISPR or blocking Ab). This could be addressed as a limitation and future direction (unless others call for and it is feasible and desired to perform additional experimentation).

Reviewer #3 (Remarks to the Author):

In this manuscript, Pronier and colleagues present a significant body of work to explore the mechanism by which Tet2 regulates the expression of MIF. The key finding is that down regulation of Tet2 results in increased expression of MIF. Mechanistically, the authors demonstrate that EGR1 drives the MIF up-regulation by binding to its promoter. The authors used primary patient samples as well as cell lines as well as primary CD34 derived monocytes in their functional studies. Overall, this is a well-conceived study with new information. there are several technical issues and additional data needed to drive home the conclusions made by the authors.

Specific comments:

1. In order to convincingly demonstrate authors need to experimentally demonstrate a) EGR1 ablation can rescue MIF up-regulation. Showing that the EGR1 ablation can prevent MIF up-regulation can also eliminate PU.1 as a confounding factor because the authors show that EGR1

site overlaps with PU.1 site in the MIF promoter.

2. The Co-IP experiments that were done to show TET2 interacts with EGR1 is of poor quality and are not convincing. It is questionable the quality/specificity of the TET2 antibody used in these Co-IP studies. considering the poor quality of the TET2 abs the authors should consider demonstrating the interaction by performing biochemical experiments. By tagging TET2 with Flag or other tag one could co-transfect tagged TET2 with EGR1 into HEK293 cells and carryout Co-IP experiments to clearly show these two proteins interact similar to the approach that was taken in a recently published paper (Jong et al., Cancer Discovery, 2019).

3. Functional experiments shown in figure 7 using differentiated monocytes. only show phenotypic data. The authors requested to show a stained micrograph of cytopins so one could appreciate morphological differences.

4. Mutation status of CMML samples are not indicated. It is important the authors indicate whether the TET2 mutations were truncation mutations, missense mutations etc. If the data are presented so missense and truncated mutations are shown as separate cohorts it will be useful.

5. Supplemental figure 2 has labeling issues. Fig. 2C and 3D are not mentioned in the manuscript.

6. TET2 knock down levels as well as flow plots for differentiation experiments should be shown.

Our response to Reviewer #1

Comment from the reviewer: The manuscript entitled “Macrophage migration inhibitory factor is a deregulated cytokine through EGR1 in resting TET2-truncating mutated monocytes” identifies an inverse association between TET2 and the cytokine migration inhibitory factor (MIF). The authors contend that loss of Tet2 enables its binding partner EGR1 to bind to the MIF promoter and facilitate its elevation. They further propose that elevated levels of MIF may be responsible for increased monocyte differentiation observed in TET2-deficient diseases, such as chronic myelomonocytic leukemia (CMML) and clonal hematopoiesis of indeterminate potential (CHIP).

A significant strength of this study is the very careful demonstration of MIF RNA and protein elevation in both mouse and human models. The authors showed MIF elevation both in vitro and in vivo and via multiple methods, including cytokine array, qRT-PCR, and Western blotting. However, the leukemia cell lines used were mostly derived from acute megakaryocytic leukemia and erythroleukemia. As the authors emphasize the role of myeloid cells in the pathogenesis of TET2-associated disease, there are acute myeloid leukemia cell lines that may lend greater support to this proposed mechanism. The introduction of the manuscript is well written and provides relevant and compelling background information. The elevation of MIF represents a novel finding that may have valuable implications for the role of inflammation in TET2-associated diseases.

Our response: We thank the reviewer for these positive comments and his/her suggestions that we carefully considered, as detailed below.

Comment from the reviewer: Weaknesses of the study include statistical considerations, minor grammatical and formatting errors, and data interpretation. Several of the human studies consist of very small numbers of patients. The translational impact of this study would be increased by the inclusion of more patients, especially considering the low penetrance, high latency, and significant variability inherent to TET2-associated diseases.

Our response: We thank the reviewer for his/her precise analysis of our paper. Following the recommendation to include more patients in translational analyses, we increased the number of CMML patients in which RT-qPCR analysis of *MIF* gene expression was performed. The new cohort of 146 (instead of 95) CMML patients includes

- 56 (instead of 35) TET2^{WT}
- 90 TET2^{MUT}, i.e., 68 (instead of 46) truncating (tr) and 22 (instead of 14) non-truncating TET2^{MUT}

We provide below two modified panels shown in revised Figure 2. This extension of the cohort validates our previous report, i.e., *MIF* gene expression is higher in monocytes of patients with a truncating *TET2* variant. Of note, this extended analysis also confirmed the correlation between truncating *TET2* variants and a decreased number of circulating platelets (see below the revised supplemental figure 2).

The revised Supplementary Figure 2 is the following :

Pronier E, Imanci A *et al.*, Supplementary Figure 2

Supplementary Table 2: Independent cohort of CMML patients used for RT-qPCR analysis of *MIF* gene expression in sorted peripheral blood monocytes.

CMML	TET2 wild type	TET2 mutated
Number of patients	56	90
Mean age in years (range)	68.9 (30-87)	73.3 (55-90)
Sex ratio M/F	26/20	61/29
CMML 0/1/2	20/18/9	34/39/11
Proliferative/Dysplastic	23/30	39/49
Leucocytes, mean. 10^9 /L (range)	21.5 (3.6-119)	20.6 (3.0-141.9)
Monocytes, mean. 10^9 /L (range)	4.8 (1.0-28.8)	5.5 (1.0-59.7)
Platelet count, mean. 10^9 /L (range)	185.5 (18-560)	124.1 (13-553)
Hemoglobin level, mean g/dL (range)	11.3 (6.3-17.2)	12.1 (6.0-16.4)
Karyotype (N/A/ND)	25/12/19	54/14/22
Mutations (mutated/analyzed)		
SRSF2	11/55 (20.0%)	42/88 (47.7%)
ASXL1	16/56 (28.6%)	31/90 (34.4%)
RUNX1	7/56 (12.5%)	15/90 (16.7%)
NRAS	9/56 (16.1%)	14/90 (15.5%)
KRAS	7/56 (12.5%)	15/90 (16.7%)
CBL	5/56 (8.9%)	12/90 (13.3%)

Comment from the reviewer: The graphs quantifying the Western blots in Figure 1E do not have error bars.

Our response: We apologize if we have not been clear enough. These graphs show the quantification of the immunoblot shown above (middle of each band normalized to Actin), which explains why there is no error bar. We considered that the reproducibility from one cell line to another (n=4) was a stronger validation than repeating immunoblots in a given cell line.

Comment from the reviewer: In Figure 1, it is surprising that elevated levels of IL-1 β and IL-6 were not detected in the cytokine array. Previous studies indicate that Tet2 loss leads to the elevation of IL-1 β secretion from proinflammatory macrophages in a mouse model of CHIP-associated cardiovascular disease. Furthermore, elevated levels of IL-1 and IL-6 have been detected in patients with CHIP-associated mutations in TET2. In addition, the authors identified three cytokines that were elevated in the absence of TET2, but only MIF was affected by altering Tet2 expression. This observation suggests that TET2 may modulate inflammatory factors via direct and indirect mechanisms; however, this notion was not addressed.

Our response: We thank the reviewer for this important suggestion. TET2 was shown to mediate *IL6* gene down regulation at the recovery phase of macrophage activation in mice (Zhang Q et al, Nature 2015), while TET2-mediated regulation of *MIF* gene expression is shown here to be detected in resting cells, as underlined in the manuscript title. The reviewer's comment is now addressed in the discussion as follows: **“Importantly, TET2 regulates *MIF* gene expression in resting monocytes while TET2-mediated *IL6* regulation was detected at the recovery phase of macrophage activation.¹³**

MIF is the main cytokine overproduced by resting monocytes in which TET2 expression is decreased, suggesting that overproduction of other cytokines by TET2-mutated cells may involve less direct effects of TET2 mutation.” (page 12)

Comment from the reviewer: In Figure 2C, the significance of including young and old healthy individuals is unclear; however, this information may be relevant to interpretations regarding CHIP.

Our response: We thank the reviewer for this question. CMML is a chronic myeloid malignancy that is mostly observed in the elderly with a median age of 72 years at diagnosis. This is the reason why, in all our studies, we explore the differential role of ageing (by comparing young and aged healthy donors) and disease (by comparing diseased cells to age-matched healthy donors). We added a statement in the text to justify these comparisons. “CMML is a clonal disorder, mostly observed in the elderly with a median age at diagnosis of 73 years.” (page 5)

Comment from the reviewer: In addition, the authors did not specify if healthy individuals in their studies were screened for CHIP-associated mutations, including those in TET2. This information could have significant implications for the interpretation of this data.

Our response: We used a 41-gene myeloid panel and NGS analysis at a mean depth > 1,000X to check the lack of CHIP in healthy donor cells studied by bulk RNA sequencing shown in Figure 2d (RNA seq analyses) and did not detect any CHIP at VAF > 2%. In the other cases, we did not have enough material to systematically screen for a CHIP. We cannot rule out that the unique age-matched healthy donor with high MIF gene expression (Figure 2c) has a TET2-mutated CHIP. We would like to argue that the probability of having more CHIP with truncating TET2 variant among the 27 healthy donors is very low.

Comment from the reviewer: In Figure 2D, the authors conclude that the expression of cytokine-encoding genes other than MIF, including IL-1 β , are not altered in CMML patients. However, the heat map shows that the expression of a variety of other cytokine-encoding genes. Consistent with previous studies, this result supports the complexity and heterogeneity of the inflammatory response in patients carrying TET2 mutations.

Our response: We agree that our statement was confusing. As already discussed, TET2 may regulate cytokine synthesis and secretion by both direct and indirect mechanisms. Our experiments identified that the direct regulation of MIF gene expression was dependent on TET2 expression level, *i.e.*, MIF gene expression was constantly increased monocytes of patients with a truncating TET2 variant. This does not exclude an indirect effect of TET2 or additional factors on the expression of other cytokine genes, in resting or activated cells, but 1) this effect is highly heterogeneous among patients, and 2) similar heterogeneity is observed in wildtype CMML cells. Importantly, mechanisms other than truncating mutations could decrease TET2 expression in CMML cells, which may explain the increased expression of MIF gene in some TET2 wildtype samples (which we checked in our RNA sequencing data).

Manuscript: “i.e. *MIF* gene expression was frequently increased in cells expressing truncating *TET2* variant whereas the increased expression of other cytokine-encoding genes was heterogeneous and independent of *TET2* status”. (page 6)

Comment from the reviewer: The lack of a clear relationship between MIF and other inflammatory signaling molecules in this experiment may actually diminish the clinical significance of MIF as it is unclear how MIF is affecting the inflammatory response. In addition, the authors demonstrate recruitment of EGR1 to genes involved in ribosome biogenesis, translation, and RNA processing, suggesting that inflammation may not be the key process affected by EGR1 and MIF. Furthermore, they do not address the consequences of elevated MIF signaling. Based on the functions of MIF and the proinflammatory environment in Tet2-deficient cells, it would be expected that elevation of MIF would enhance or contribute to the proinflammatory environment promoted by Tet2 loss; however, the authors do not demonstrate or address this possibility. Additional experiments, clarification of this mechanism, and discussion of these notions would enhance the impact of this study. The results in Figure 7 are not as convincing, showing a weak correlation and not demonstrating causation. However, the interpretation of these results is vital to the overall hypothesis that the elevation of MIF in the absence of TET2 drives enhanced myeloid differentiation. This data could be considerably strengthened by additional experiments. In general, the discussion section is less relevant to the current study and could be restructured. The authors consistently emphasize a potential role for MIF in cardiovascular disease. While compelling, this connection was not addressed in the current study.

Our response: We thank the reviewer for his/her suggestions. To strengthen the results initially shown on Figure 7, we performed additional experiments. Cord blood CD34⁺ cells were cultured in medium with SCF, FLT3L, IL-3 and G-CSF in the presence or absence of recombinant MIF (20 ng/ml) for 48h before bulk or single-cell RNA-seq, which is now shown on the new supplementary Figure 8 (bulk) and Figure 7 and new supplementary figure 9 and 10 (scRNAseq).

Pronier E, Imanci A *et al.*, Supplementary Figure 8

Manuscript: “Cord blood CD34⁺ cells were cultured with SCF, FLT3L, IL-3 and G-CSF in the presence or absence of 20 ng/mL recombinant MIF for 48 hours before bulk RNA-seq analysis: 33 genes were differentially expressed in the presence of MIF (**Supplementary Fig. 8a**). GO Molecular Function analysis of differentially deregulated genes (DEGs) identified kinase activity (p value=3.44e⁻⁶; FDR q value= 2.94e⁻³) (**Supplementary Fig. 8b**) and signaling receptor binding (p value=3.34e⁻⁵; FDR q value= 1.43e²) (**Supplementary Fig. 8c**) signatures. GO Biological Process analysis of DEGs identified cell morphogenesis involved in differentiation (p value=3.33e⁻⁶; FDR q value= 6.23e⁻³) (**Supplementary Fig. 8d**) and anatomical structure formation involved in morphogenesis (p value=4.43e⁻⁵; FDR q value= 2.76e⁻²) (**Supplementary Fig. 8e**) signatures. Finally, CIBERSORT analysis identified more monocytes and anti-inflammatory macrophages in the culture when MIF was added (**Supplementary Fig. 8f**), further suggesting that MIF may promote the monocytic differentiation of stem and progenitor cells.” (Page 10)

“Single cell analysis of cells collected at day 7 showed 7 clusters defined by the expression of characteristic genes (**Supplementary Fig. 9a**) and further validated by the ten most expressed genes (**Supplementary Fig. 9b**). MIF induced an increase in the number of cells in cluster 6 (monocytes) while this number was decreased in cluster 4 (granulocytes; **Fig. 7e**). Accordingly, MIF increased the expression in *CSF1R* gene, a SPI1/PU.1 transcription factor target, in cluster 6 while decreasing the expression of *GFI1* gene in cluster 4 (**Fig. 7f**). These results further argue for the ability of MIF to promote the expansion of monocytes at the expense of granulocytes (**Fig. 7g**).” (Page 10)

New data in Figure 7

New Supplementary Figure 9

Pronier E, Imanci A *et al.*,
Supplementary Figure 9

Comment from the reviewer: The innovation of this study is the identification of a novel cytokine that is associated with TET2 mutations. While previous studies indicate a role for TET2 in modulating inflammation, MIF has not been detected, and these studies often employed bacterial challenge experiments. The significance of this study is that this new mechanism for TET2-associated disease may represent a novel therapeutic target for patients with these mutations. A unique aspect of this study is the detection of elevated MIF under resting or unchallenged conditions, providing support for an intrinsic role for TET2 in the inflammatory response. Collectively, this manuscript presents interesting novel findings that may have significant translational impact. It is recommended that the authors address the experimental concerns about the manuscript, clarify the mechanism and data interpretation, and correct grammar, typos, and formatting.

Our response: We thank the reviewers for his/her useful comments that helped us to improve the proposed manuscript.

Our response to Reviewer #2

Comment from the reviewer: Inactivating TET2 mutations are among the most common drivers of clonal hematopoiesis and myeloid neoplasms, especially chronic myelomonocytic leukemia (CMML). To this point, the inflammatory sequelae of TET2 mutations (especially regarding the increased risk of cardiovascular disease) have been mainly related to increased NLRP3 inflammasome activation and IL-1b and IL-6 production by TET2-mutant monocytes/macrophages. In this interesting and timely manuscript, Dr. Pronier et al. use a host of complementary experiments in murine and human cells to demonstrate increased, EGR1-driven production of macrophage migration inhibitor factor (MIF) in TET2-deficient cells, potentially contributing to skewed monocytic differentiation. Although not explored in this manuscript, targeting MIF to prevent TET2 mutation-driven leukemogenesis and cardiovascular disease may warrant further study.

The following comments are intended to improve the quality of the manuscript.

Our response: We thank the reviewer for these positive comments and his/her careful reading of our manuscript.

Comment from the reviewer: 1. Introduction - line 6 - "activity of TET enzyme activity" - redundant.

Our response: We apologize for this error that has been corrected.

Comment from the reviewer: 2. Intro - use of "ARCH" is acceptable but the risk of cardiovascular disease has been established for clonal hematopoiesis with variant allele fraction of at least 2% (i.e. definition of "CHIP"). Please specify or use "CHIP".

Our response: We followed the reviewer suggestion and replaced ARCH by CHIP

Manuscript: "named **Clonal Hematopoiesis of Indeterminate Potential (CHIP)**.²⁷ These *TET2*-mutated clones can be a first step towards a malignancy such as CMML.²⁸ A *TET2*-mutated **CHIP** also increases the risk of death from cardiovascular disease.^{29,30} A deregulated production of inflammatory cytokines by mutated myeloid cells may explain the cardiovascular risk associated with *TET2* **CHIP** as these cytokines may promote leukocyte recruitment to atherosclerotic plaques." (page 3-4)

Comment from the reviewer: 3. Results - paragraph 1 - "increased in four human leukemic cell lines (kasumi-1, M07e, UT-7 and TF-1) in which TET2 gene expression was decreased" - please specify "using shRNA" or something to this effect. Otherwise, the impression may be given in the text that these cell lines naturally have decreased TET2 expression. On this note, it might be helpful to clarify if these cell lines contain any known TET2 mutations.

Our response: We added the suggested information: "were increased in four, TET2 wildtype human leukemic cell lines (kasumi-1, M07e, UT-7 and TF-1) in which *TET2* gene expression was decreased **by using lentiviral shRNA**, as previously described (**Supplementary Fig. 1a**)." (Page 5) Also, supplemental Figure 1a now shows the down-

regulation of *TET2* gene expression, as requested. Of note, none of the used cell lines is *TET2* mutated, which is also added in the material and methods (page 14, “**which all express wildtype *TET2***”)

Comment from the reviewer: 4. Results - paragraph 1 - "excluding differential monocytes counts as a confounding factor" - please specific if differential counts were only assessed in blood, or also in bone marrow.

Our response: We now specify that differential counts were only assessed in the peripheral blood. "excluding differential monocyte counts **in the peripheral blood** as a confounding factor" (Page 5)

Comment from the reviewer: 5. Results - p. 6 & Suppl. Fig. 2A/B - please explain why n=50 total *TET2*-mut in Suppl. Fig 2A but n = 38+19 in 2B.

Our response: We apologize for this error. Following a recommendation of reviewer#1, we have increased the number of patients in which RT-qPCR analysis of *MIF* gene expression was performed. The new cohort of 146 CMML patients includes

- 56 *TET2*^{WT}
- 90 *TET2*^{MUT}, i.e., 68 truncating (tr) and 22 non-truncating *TET2*^{MUT}

We provide two modified panels (c and f) in revised Figure 2, a new supplementary Figure 2 and a new supplementary Table 2, and paid attention to the numbers in this revised version (see more details in our response to reviewer # 1). (Page 6)

Comment from the reviewer: 6. Results - p. 7 and Fig. 3c - can the authors please clarify - it seems counter-intuitive that decrease of H3K4me3 (normally associated with transcriptional activation) in cells treated with *TET2* shRNA would be associated with increased *MIF* promoter activity and transcription. Am I misunderstanding?

Our response: We confirm that we observed a decrease of H3K4me3 (normally associated with transcriptional activation) in cells in which *TET2* has been down regulated (shRNA) as well as in *TET2* mutated CMML monocytes and we agree with the reviewer that this observation could be seen as counter-intuitive.

Actually, such an observation was already reported, *e.g.* reduced H3K4me3 marks is decreased in cytokine and chemokine transcriptionally active genes during dendritic cell differentiation (Huang et al., *Genes Immun.*, 2012). In addition, our group was involved in a previous study (Deplus et al., *EMBO J*, 2013) showing that TET2, together with TET3 and OGT, co-localized at CpG islands and transcription start sites where they influence H3K4 trimethylation of active promoters, *i.e.*, TET2 depletion decreases H3K4me3. Accordingly, in Kasumi cells in which *TET2* was down regulated, we observed a strong diminution of H3K4me3 marks at the TSS region of *MIF* promoter while active polymerase II was more recruited. As an explanation, we may miss other histone marks that compensate the decreased H3K4me3.

Comment from the reviewer: 7. Results - p. 8, Fig. 6 & Suppl Figs. 6 - for the examples of EGR1 distribution around TSS for *RAD50* and other genes listed, presumably these are relevant to the subsequent Gene Ontology analysis? If so, it may strengthen their selection to clarify. Can other factors (*i.e.* non-TET2 mutations) in CMML affect EGR1 peak localization?

Our response: Actually, the chosen genes to show EGR1 distribution around TSS, such as *RAD50*, are part of the genes used for Gene Ontology analysis, which was clarified in the text. We agree that other factors could potentially affect EGR1 peak localization. We could not check any hypothesis as we did not have enough material to extend ChIPseq analyses to other potential factors. “Gene Ontology (GO) analysis of EGR1-interacting genes, using peaks that are common to healthy donor and CMML monocytes,” (Page 9)

Comment from the reviewer: 8. Results - p. 9 - why was *MIF* expression not compared in transcriptomic data of CD34 cells from healthy donor and CMML patients (+/- TET2 mutations)? Is it not expressed in CD34+ cells? Was it not covered in the array?

Our response: We performed this analysis and did not show the results nor added any comment as *MIF* gene expression was similar in healthy donor and CMML CD34+ cells, indicating that the deregulation of *MIF* gene was specific of mature monocytes. Together with the influence of *MIF* on CD34+ cell differentiation, it suggests a regulatory loop between monocytes in which *MIF* gene is overexpressed and CD34+ cells whose differentiation into monocytes is enhanced by *MIF*.

Comment from the reviewer: 9. Discussion - p. 11 & elsewhere - re: "in the absence of infectious or inflammatory stimulus" - can the residual *ex vivo* effects of possible *in vivo* exposure to such a stimulus be completely excluded in primary patient monocytes? Can epigenetic effects of endogenous exposures be excluded?

Our response: Thank you for this comment. Cells are not collected in the setting of active infection. Nevertheless, multiple and heterogeneous inflammatory cytokines are detected in the plasma and the bone marrow of CMML patients (Niyongere S, et al, Leukemia 2019) and could generate epigenetic effects. We meant that the cells were not exposed to an additional *ex vivo* strong inflammatory stimulus such as LPS or IL-6. Therefore, we added the word "additional" to the mentioned sentence. (Page 12)

Comment from the reviewer: 10. Discussion - p. 12 - what is the significance of amino acids 86-102 of MIF? It may be helpful to describe the structure/function of MIF in the Introduction or the relevance of this region in the Discussion or to exclude this detail.

Our response: We removed this useless information and apologize for the confusing message.

Comment from the reviewer: 11. Methods - p. 13 - what is the source of SCF, IL-3, IL-6, TPO-FLT3? i.e. vendor, species, recombinant?

Our response: We have now added this information to our material and method section as follows: "stem cell factor recombinant (SCF, 50 ng/mL; Immunex), interleukin-3 recombinant (IL-3, 10 ng/mL; Novartis), IL-6 recombinant (10 ng/mL; Peprotech), thrombopoietin recombinant (TPO, 10 ng/mL; Peprotech), Fms-like tyrosine kinase 3 recombinant (FLT-3, 50 ng/mL; Diaclone), granulocyte colony-stimulating factor recombinant (G-CSF 10 ng/mL; Peprotech), in a 37°C incubator with 5% CO₂." (Page 14)

Comment from the reviewer: 12. Fig. 6 title is followed by "(a,f)". What does this indicate?

Our response: We thanks the reviewer for his/her careful reading. This is a typo error that has been corrected in the new version.

Comment from the reviewer: 13. Fig. 1a - numbering of columns stops at 12, when 20 are present.

Our response: We have now fixed this error.

Comment from the reviewer: 14. Am I missing the data proving that shTET2 knocks down TET2 expression at mRNA and/or protein level?

Our response: We have added these data in supplementary Figure 1a.

Comment from the reviewer: 15. Fig. 2 - can the authors discuss why some healthy donors and TET2-WT CMML patients have high expression of MIF? And why do TET2-MUT CMML patients not uniformly over-express MIF? This doesn't seem to be fully explained by truncating vs non-truncating mutations. What about confounding pathways? Is it possible other mutations (non-TET2) could increase MIF or interfere with tr-TET2-mutations?

Our response: We totally agree with this comment. Many parameters appear to influence the level of *TET2* gene expression. In CMML patients with *TET2* mutation, truncation of the proteins and the VAF of the mutated allele appear to be key parameters. In patients with *TET2* WT CMML and in healthy donors, *TET2* gene expression is also heterogeneous and the mechanisms involved are unclear. This is now discussed in the revised manuscript.

“Many parameters may influence *TET2* gene expression. In CMML patients with a *TET2* variant, truncation of the protein and the VAF of the mutated allele play essential roles. *TET2* gene expression can also be decreased in the absence of gene mutation through poorly understood mechanisms.” (Page 12)

Comment from the reviewer: 16. Fig. 2D - a selection of cytokines & chemokines are shown. Are the full list of targets described somewhere in the manuscript?

Our response: We have added a new supplementary data 1 sheet with all the normalized count per sample (Supplementary Data 1_RNAseq_normalized_counts).

Comment from the reviewer: 17. Fig. 2G - it is understood why bone marrow plasma is shown but have the authors also compared MIF expression in blood plasma or serum?

Our response: Yes, in addition to peripheral blood plasma, we have analyzed MIF protein level in peripheral blood serum. We performed cytokine arrays in 3 healthy donors and 3 truncating TET2^{MUT} CMML patients, again showing more MIF in the serum of trTET2^{MUT} CMML patients. As the number of cases is low, this information is shown as a supplemental figure (panel 3d).

Comment from the reviewer: 18. Fig. 5 a and b - were insufficient cells available to run CMML samples in parallel in parts a and b? Only CMML 1900 overlaps in both panels.

Our response: Indeed, the number of cross-linked cells was not sufficient to run all the samples shown on Fig. 5a and b in parallel.

Comment from the reviewer: 19. Fig.7 & related manuscript section - the authors cite a reference that MIF can increase SPI1/PU.1 but don't close the loop here (showing either increased SPI1 or decreased GFI1 in this culture system). Similarly, they talk about a feedback loop but don't try to interfere with MIF (or EGR1) in their experiments (i.e. CRISPR or blocking Ab). This could be addressed as a limitation and future direction (unless others call for and it is feasible and desired to perform additional experimentation).

Our response: We agree with the reviewer's comment that this important point needed to be demonstrated (which was requested also by reviewer 3). We used siRNA targeting EGR1 (3 sets) that were introduced in isolated monocytes from 2 trTET2^{MUT} CMML patients before studying *EGR1* and *MIF* expression by RT-qPCR. These data are now included in the manuscript as Figure 5b (normalizer, *RPL32*) and supplemental figure 5a (normalizer, *HPRT*) and show that EGR1 down regulation decreases *MIF* gene expression.

Manuscript: "To demonstrate EGR1-dependent *MIF* expression in trTET2^{MUT} cells, we transfected EGR1-siRNA in trTET2^{MUT} monocytes collected from 2 CMML patients. *EGR1* mRNA down regulation, validated by RT-qPCR (Fig. 5b and Supplementary Fig. 5a), prevented *MIF* up-regulation in these cells." (Page 8)

Supplementary Figure 5a

Our response to Reviewer #3

Comment from the reviewer: In this manuscript, Pronier and colleagues present a significant body of work to explore the mechanism by which Tet2 regulates the expression of MIF. The key finding is that down regulation of Tet2 results in increased expression of MIF. Mechanistically, the authors demonstrate that EGR1 drives the MIF up-regulation by binding to its promoter. The authors used primary patient samples as well as cell lines as well as primary CD34 derived monocytes in their functional studies. Overall, this is a well-conceived study with new information. there are several technical issues and additional data needed to drive home the conclusions made by the authors.

Specific comments:

1. In order to convincingly demonstrate authors need to experimentally demonstrate a) EGR1 ablation can rescue MIF up-regulation. Showing that the EGR1 ablation can prevent MIF up-regulation can also eliminate PU.1 as a confounding factor because the authors show that EGR1 site overlaps with PU.1 site in the MIF promoter.

Our response: We agree with the reviewer's comment that this important point needed to be demonstrated, which was requested also by reviewer 2. We used siRNA targeting EGR1 (3 sets) that were introduced in monocytes sorted from 2 trTET2^{MUT} CMML patients before studying *EGR1* and *MIF* expression by RT-qPCR. These data are now included in the manuscript as Figure 5b (normalizer, *RPL32*) and supplemental figure 5a (normalizer, *HPRT*) and show that EGR1 down regulation decreases *MIF* gene expression.

Manuscript: "To demonstrate EGR1-dependent *MIF* expression in trTET2^{MUT} cells, we transfected EGR1-siRNA in trTET2^{MUT} monocytes collected from 2 CMML patients. *EGR1* mRNA down regulation, validated by RT-qPCR (Fig. 5b and Supplementary Fig. 5a), prevented *MIF* up-regulation in these cells." (Page 8)

Of note, EGR1 site overlaps with SP1 (specificity protein 1, also known as Transcription factor Sp1) but not with SPI1/PU.1 site from the ETS-domain transcription factor family.

Comment from the reviewer: 2. The Co-IP experiments that were done to show TET2 interacts with EGR1 is of poor quality and are not convincing. It is questionable the quality/specificity of the TET2 antibody used in these Co-IP studies. considering the poor quality of the TET2 abs the authors should consider demonstrating the interaction by performing biochemical experiments. By tagging TET2 with Flag or other tag one could co-transfect tagged TET2 with EGR1 into HEK293 cells and carryout Co-IP experiments to clearly show these two proteins interact similar to the approach that was taken in a recently published paper (Jong et al., Cancer Discovery, 2019).

Our response: We agree that due to the poor quality of available TET2 antibodies, these data were not convincing. We have now co-transfected pcDNA3-HA-TET2 and pcDNA3-EGR1 into 293T cells and have performed co-immunoprecipitation using either anti-HA or anti-EGR1 antibodies. Previous data are now shown as supplementary Figure 5b & 5c and we added these new data as Figure 5d and 5e.

Manuscript: “To determine if EGR1 and TET2 could interact, we transiently co-transfected 293T cells with empty vectors or pcDNA3-EGR1 and pcDNA3-TET2-HA. Having checked EGR1 and TET2-HA protein expression (Fig. 5d), co-immunoprecipitation experiments validated the ability of TET2 to interact with EGR1. Interestingly, HDAC1 was only immunoprecipitated with TET2-HA antibody, suggesting a stronger interaction with TET2 in the complex (Fig. 5d). Co-immunoprecipitation experiments in human blood monocytes confirmed TET2 interaction with EGR1 (Supplementary Fig. 5b and 5c) and validated the ability of TET2 to interact with HDAC1 and HDAC2 (Supplementary Fig. 5d).” (Page 8)

Comment from the reviewer: 3. Functional experiments shown in figure 7 using differentiated monocytes. only show phenotypic data. The authors requested to show a stained micrograph of cytopins so one could appreciate morphological differences.

Our response: As granulocytes are very fragile, cytopins showed many dead cells. Therefore, to answer the reviewer’s comment, we performed additional experiences.

Cord blood CD34⁺ cells were cultured in medium with SCF, FLT3L, IL-3 and G-CSF in the presence or absence of 20 ng/mL recombinant MIF for 48h before bulk and single-cell RNA-sequencing, which is now shown on the new supplementary Figure 8 (bulk) and Figure 7 and supplementary Figure 9 (scRNAseq).

Pronier E, Imanci A *et al.*, Supplementary Figure 8

Manuscript: “Cord blood CD34⁺ cells were cultured with SCF, FLT3L, IL-3 and G-CSF in the presence or absence of 20 ng/mL recombinant MIF for 48 hours before bulk RNA-seq analysis: 33 genes were differentially expressed in the presence of MIF (**Supplementary Fig. 8a**). GO Molecular Function analysis of these differentially deregulated genes (DEGs) identified kinase activity (p value=3.44e⁻⁶; FDR q value= 2.94e⁻³) (**Supplementary Fig. 8b**) and signaling receptor binding (p value=3.34e⁻⁵; FDR q value= 1.43e²) (**Supplementary Fig. 8c**) signatures. GO Biological Process analysis of these DEGs identified cell morphogenesis involved in differentiation (p value=3.33e⁻⁶; FDR q value= 6.23e⁻³) (**Supplementary Fig. 8d**) and anatomical structure formation involved in morphogenesis (p value=4.43e⁻⁵; FDR q value= 2.76e⁻²) (**Supplementary Fig. 8e**) signatures. Finally, CIBERSORT analysis identified more monocytes and anti-inflammatory macrophages in culture when MIF was added (**Supplementary Fig. 8f**), further suggesting that MIF may promote the monocytic differentiation of stem and progenitor cells.” (Page 10)

“We then performed single cell analysis at day 7, identifying 7 clusters whose identification was based on the expression of characteristic genes (**Supplementary Fig. 9a**), which was validated by analyzing the 10 most expressed genes (**Supplementary Fig. 9b**). MIF induces an increase in the number of cells in cluster 6 while this number was decreased in cluster 4 (**Fig. 7e**). Accordingly, MIF increased the expression in *CSF1R* gene, a SPI1/PU.1 transcription

factor target, in cluster 6 while decreasing the expression of *GF11* gene in cluster 4 (Fig. 7f). Validating our observation performed by flow cytometry analysis, MIF promoted the expansion of monocytes at the expense of granulocytes (Fig. 7g).” (Page 10-11)

New data in Figure 7

New Supplementary Figure 9

Pronier E, Imanci A et al.,
Supplementary Figure 9

Comment from the reviewer: 4. Mutation status of CMML samples are not indicated. It is important the authors indicate whether the *TET2* mutations were truncation mutations, missense mutations etc. If the data are presented so missense and truncated mutations are shown as separate cohorts it will be useful.

Our response: This is an important comment and we have now indicated in the new version of the manuscript the status of *TET2* mutations. We have also analyzed bone marrow MIF according to *TET2* status.

Figure 2 g

Figure 5a

Figure 5c

Comment from the reviewer: 5. Supplemental figure 2 has labeling issues. Fig. 2C and 3D are not mentioned in the manuscript.

Our response: We apologize for these mistakes. We have now mentioned these figures and supplementary Figure 2c was labeled appropriately.

Manuscript: “No difference in white blood cell, monocyte, neutrophil, lymphocyte count and hemoglobin level were observed between the 3 groups. Interestingly, the platelet count was significantly lower in trTET2^{MUT} patient subgroup (Supplementary Fig. 2c).” (Page 6)

“MIF level was higher in the supernatant (18 hours in serum-free medium) of trTET2^{MUT} CMML-monocytes from 2 patients compared to healthy donors (Supplementary Fig. 3b, 3c) and in the serum of 3 trTET2^{MUT} CMML patients at diagnosis compared to 3 healthy donor serum samples (Supplementary Fig. 3b, 3d).”

Comment from the reviewer: 6. TET2 knock down levels as well as flow plots for differentiation experiments should be shown.

Our response: We have now added data proving that shTET2 diminishes TET2 gene expression at mRNA level in supplementary Figure 1a.

We have also added the gating strategy as **Supplementary Figure 10**.

Pronier E, Imanci A *et al.*, Supplementary Figure 10

REVIEWERS' COMMENTS:

Reviewer #1 (Remarks to the Author):

Authors have addressed all of my comments in a satisfactory manner. This revised manuscript is significantly improved.

Reviewer #2 (Remarks to the Author):

The authors have adequately addressed my concerns. I congratulate them on this study.